# Nutritional and host environments determine community ecology and keystone species in a synthetic gut bacterial community

Anna S. Weiss [1], Lisa S. Niedermeier[1], Alexandra von Strempel[1], Anna G. Burrichter[1], Diana Ring[1], Chen Meng[2], Karin Kleigrewe [2], Chiara Lincetto[3], Johannes Hübner [3] & Bärbel Stecher[1,4] ✉

A challenging task to understand health and disease-related microbiome signatures is to move beyond descriptive community-level profiling towards disentangling microbial interaction networks. Using a synthetic gut bacterial community, we aimed to study the role of individual members in community assembly, identify putative keystone species and test their influence across different environments. Single-species dropout experiments reveal that bacterial strain relationships strongly vary not only in different regions of the murine gut, but also across several standard culture media. Mechanisms involved in environment-dependent keystone functions in vitro include exclusive access to polysaccharides as well as bacteriocin production. Further, *Bacteroides caecimuris* and *Blautia coccoides* are found to play keystone roles in gnotobiotic mice by impacting community composition, the metabolic landscape and inflammatory responses. In summary, the presented study highlights the strong interdependency between bacterial community ecology and the biotic and abiotic environment. These results question the concept of universally valid keystone species in the gastrointestinal ecosystem and underline the context-dependency of both, keystone functions and bacterial interaction networks.

Intestinal microbial communities are of key importance for their mammalian host. A current challenge in microbiome research is to understand the functional relevance of particular community configurations or "signatures", associated with health and disease states[1]. Gut microbial community composition is influenced by a variety of abiotic and biotic factors, including temperature, pH, diet, host immune defenses, metabolites, and microbe-microbe interactions[2–6].

Interactions among individual bacteria can be cooperative, neutral, or exploitative, involving, e.g. nutrient degradation, exchange of metabolites, and production of inhibitory compounds. Overall, this results in a complex trophic bacterial network[7,8]. Several studies identified specific bacteria to be of key importance for the overall structure of intestinal trophic networks[9]. The keystone species concept, initially developed for animal ecology[10] and later adapted to microbial ecology, refers to taxa

[1]Max von Pettenkofer Institute of Hygiene and Medical Microbiology, Faculty of Medicine, LMU Munich, Munich, Germany. [2]Bavarian Center for Biomolecular Mass Spectrometry, TUM School of Life Sciences, Technical University of Munich, Freising, Germany. [3]Division of Paediatric Infectious Diseases, Dr. von Hauner Children's Hospital, Ludwig Maximilians University, Munich, Germany. [4]German Center for Infection Research (DZIF), partner site LMU Munich, Munich, Germany. ✉e-mail: stecher@mvp.lmu.de

that take on a crucial role in the ecosystem[11]. Microbial keystone taxa are described as highly connected taxa, and when lacking, result in changes in microbiome composition and/or function at a particular space or time[12]. These taxa often, but not always, have an over-proportional influence in the community, relative to their abundance[11]. In the context of the mammalian gut microbiome, species providing important metabolic functions such as degrading starch or polysaccharides[13–16] or producing short-chain fatty acids[17], as well as species being associated with anti-inflammatory responses[18], restoring dysbiosis and promoting gut health[19,20] have been termed keystones. Due to their important role in the ecosystem, identifying and targeting such keystone taxa may open new entry points for microbiome-targeted therapies. However, given the complexity of the gastrointestinal ecosystem as well as the inter-individual differences between gut microbial communities, the generality and universal applicability of the keystone species concept in the context of the gut microbiome remains elusive.

High-throughput sequencing and metabolomics technologies have generated a wealth of data, which can be exploited by modeling approaches to shed light on the underlying processes of potential keystone species shaping the microbiome[21–23]. In particular, inference of microbial interaction and co-occurrence networks are powerful tools for delineating microbial community structures[24,25]. While the computationally identified bacterial associations may result from true ecological relationships, they cannot be distinguished from associations occurring due to environmental selection[26] and the biological interpretation often remains uncertain and calls for experimental validation[27,28]. For this purpose, synthetic model communities are excellent tools, as community members are well-characterized, interactions can be experimentally determined and hypotheses can be verified in a systematic way[29–32]. These experimental model systems also enable the identification of community members with special importance for the ecosystem, i.e., keystones in Paine's sense[10], by allowing the implementation of systematic presence-absence studies[30,33].

In this study, we employed the Oligo-Mouse-Microbiota (OMM[12]), a widely used synthetic bacterial model community, which can be studied in vitro and in gnotobiotic mice[34–36]. The OMM[12] model is publicly available, adaptable, and stable[37]. In addition, it recapitulates important phenotypes of a complex microbiota in gnotobiotic mice, including colonization resistance to pathogens[38] and immune development[39]. Using this well-established model, we set out to study the role of individual members in community assembly, to identify putative keystone species and test if their influence is universal across different environments. We generated dropout communities by systematically removing one species at a time and compared community ecology and species relationships across different in vitro cultivation media as well as physiologically different regions of the murine gut.

Our study revealed that the observed bacterial interactions as well as identified keystone species' impact are rarely conserved across different commonly used culture media and also differ among different regions of the murine gut. This questions the concept of universally valid keystone species in the gastrointestinal ecosystem and suggests that both, keystone functions and bacterial interactions are strongly context-dependent.

## Results

### The nutritional environment configures community assembly in vitro

As a first step, we analyzed community assembly of the OMM[12] consortium in different culture media using an in vitro batch culture approach[35]. The OMM[12] is a synthetic bacterial model community that harbors twelve bacteria representing the five major eubacterial phyla of the murine gut microbiota (Table 1).

Bacterial communities were stabilized in five different culture media for four days with serial dilutions every 24 h (Fig. 1a, adaptations to the previously published protocol as described in Fig. S1). To analyze the resulting community composition, the absolute abundance of the individual species at day four of cultivation was determined by qPCR as normalized 16S rRNA copies per ml culture. The chosen culture media differ in both, the supplied carbon sources and the complexity of the medium basis (Table 2, Table S1). Of note, AF and APF medium only differ in the supplied carbohydrate sources but share the same complex medium basis (Table S1).

The community composition at stabilization differed distinctly across the five conditions. This confirmed that both, the supplied carbon sources with fixed background medium composition (e.g., AF vs. APF medium) as well as the supplied background medium composition with fixed carbohydrate source (e.g., glucose in AF vs. TYG medium) strongly affect community assembly (Fig. 1b, c, Fig. S1A). Correspondingly, the pH of the spent culture supernatant of the community cultures after four days of cultivation significantly differed between the conditions, as determined by the difference between spent culture supernatant pH and fresh medium pH $\triangle pH = pH_{spent\ medium} - pH_{fresh\ medium}$ (Fig. S1B). The species richness was the highest in mGAM with ten of twelve bacteria coexisting at day four, followed by AF, APF, and YCFA medium with eight and TYG medium with only five bacteria coexisting, respectively (Fig. 1c).

### The carbohydrate environment determines key species driving community assembly in vitro

To gain insights into the role of the twelve species in community assembly and to identify keystone species, we generated dropout communities, each lacking one individual strain at a time. The

## Table 1 | Members of the synthetic OMM[12] community

| Strain | Strain ID | Letter code | Notable metabolic capabilities[34,35] |
|---|---|---|---|
| *Clostridium innocuum* I46 | DSM 26113 | *C.in* | Butyrate, valerate and caproate production |
| *Bacteroides caecimuris* I48 | DSM 26085 | *B.ca* | Polysaccharide degradation, production of branched SCFA from branched AA |
| *Limosilactobacillus reuteri* I49 | DSM 32035 | *L.re* | Lactate and acetate production |
| *Bifidobacterium longum* subsp. *animalis* YL2 | DSM 26074 | *B.an* | Lactate and acetate production |
| *Muribaculum intestinale* YL27 | DSM 28989 | *M.in* | Succinate and propionate production, AA biosynthesis |
| *Flavonifractor plautii* YL31 | DSM 26117 | *F.pl* | Butyrate production (from lysine) |
| *Enterocloster clostridioformis* YL32 | DSM 26114 | *E.cl* | Acetate production, AA biosynthesis |
| *Akkermansia muciniphila* YL44 | DSM 26109 | *A.muc* | Mucin utilization, propionate, and succinate production |
| *Turicimonas muris* YL45 | DSM 26109 | *T.mu* | Aspartate degradation, formate production |
| *Blautia coccoides* YL58 | DSM 26115 | *B.co* | Acetogenesis |
| *Enterococcus faecalis* KB1 | DSM 32036 | *E.fa* | Lactate production, conversion of arginine to ornithine |
| *Acutalibacter muris* KB18 | DSM 26090 | *A.mur* | Glutamate fermentation, AA biosynthesis |

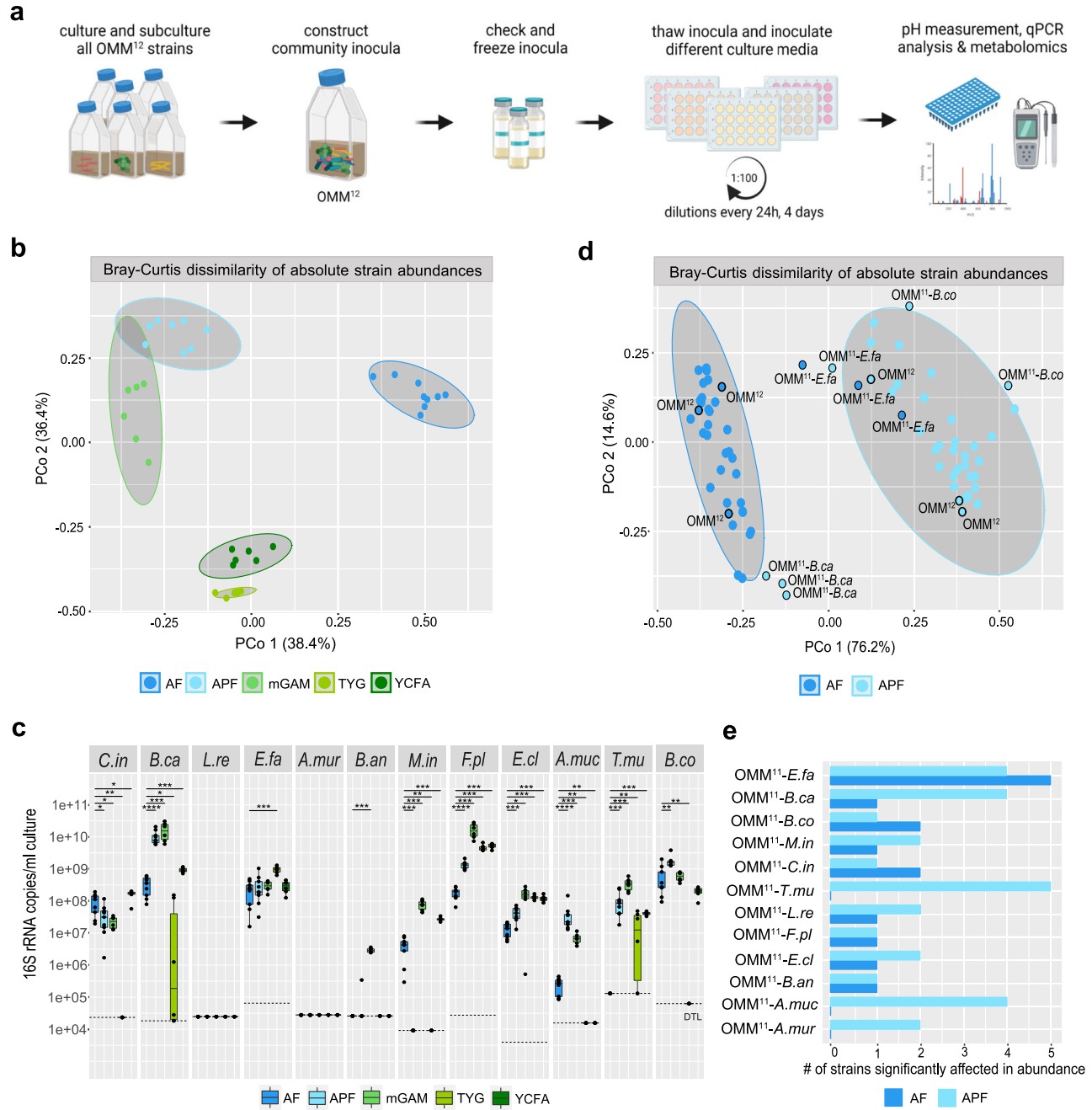

**Fig. 1 | Community assembly of the full consortium and single-species dropout communities in different media. a** Experimental workflow depicting in vitro community analysis. Bacterial monocultures were prepared and inocula were mixed. Generated community inocula were validated and frozen in glycerol stock vials. Thawed inocula were cultured in different media for four days with serial dilutions every 24 h. Culture supernatants from densely grown community cultures were sterile-filtered, samples for pH measurements and mass spectrometry were collected and the bacterial pellet was stored for qPCR analysis. **b** Principle coordinate analysis of the community structure of the OMM12 consortium in different cultivation media. Bray–Curtis dissimilarity analysis was performed on absolute abundances of the individual species ($N = 9$ for AF and APF medium, $N = 6$ for mGAM, TYG, and YCFA medium) comparing community assembly across the different culture media. Cultural media are depicted in different colors. Gray ovals cluster each culture medium with a 95% confidence interval. **c** Community composition of the full consortium in different cultivation media. Median absolute abundances (black line, $N = 9$ for AF and APF medium, $N = 6$ for mGAM, TYG, and

YCFA medium) are shown with the corresponding upper and lower percentile (box, whiskers indicate 1.5 times interquartile range, points outside this range are considered outliers) for all individual species. Significant differences between absolute abundances in AF medium and the four other media are depicted by asterisks (two-sided Wilcoxon test, $p$ values denoted as *<0.05 **<0.01, ***<0.005, ****<0.0001, ns not shown). **d** Principle coordinates analysis of community structure in AF and APF medium. Bray–Curtis dissimilarity analysis was performed on absolute abundances of the individual species (median of $N = 3$ replicates each) comparing the OMM12 and the twelve dropout communities. Cultural media are depicted in different colors. Gray ovals cluster each culture medium with a 95% confidence interval, outliers from the corresponding clusters, as well as the OMM12 community for reference, are highlighted with a black ring. **e** Bar plot quantifying the influence each strain had on community assembly by counting the number of species significantly affected in their absolute abundance in the individual dropout communities compared to the full consortium for two culture media. Source data are provided with this paper.

**Table 2 | Culture media used in this study**

| Medium | Main carbon source | Source |
|---|---|---|
| AF (Anaerobic FCS medium) | Glucose | 35 |
| APF (Anaerobic polysaccharide FCS medium) | Arabinose, fucose, lyxose, rhamnose, xylose, xylan, inulin, mucin | Adapted from[35] |
| mGAM (Modified Gifu anaerobic medium) | Glucose, starch | Commercial (HiMedia Labs) |
| TYG (Tryptone yeast extract glucose medium) | Glucose | 69 |
| YCFA (Yeast casitone fatty acids medium) | Glucose, starch, cellobiose | 70 |

previously obtained data set on full community assembly served as a reference data set to identify changes in community ecology. Community assembly of all twelve dropout communities was studied using the batch culture approach described above in two media that resulted in distinctly different community structures (Fig. 1b) but only differed in the supplied carbohydrate sources: AF medium containing glucose and APF medium containing other C5 and C6 sugars as well as polysaccharides (Table 2, Table S1).

Bray–Curtis dissimilarity analysis of the median absolute abundance of the individual species in all communities revealed clear differences (95% confidence interval) between communities grown in the glucose (AF medium) vs. the polysaccharide condition (APF medium, Fig. 1d). Notably, three dropout communities stood out from the corresponding clusters: the community lacking *E. faecalis* in glucose medium (AF) and the communities lacking *B. caecimuris* and *B. coccoides* in polysaccharide medium (APF). Comparing the absolute abundance of all species in the corresponding dropout communities to the absolute abundance in the full consortium (Fig. S2) revealed, that the absence of these species had among the strongest impact on the other species' abundances. *E. faecalis*, affected five and five species, *B. caecimuris* affected one and four species and *B. coccoides* affected two and one species in the glucose (AF) and polysaccharide medium (AFP), respectively (Fig. 1e). This suggests that depending on the carbohydrate environment, species relationships can differ distinctly and different bacterial species take on a key ecological role in driving community assembly.

**Altered community assembly is linked to distinct environmental modifications**

Analysis of pH changes ($\triangle pH = pH_{spent\ medium} - pH_{fresh\ medium}$) in the spent culture supernatant of the dropout communities revealed that communities lacking *E. faecalis* showed stronger acidification than the $OMM^{12}$ community in both media conditions (median $OMM^{11\text{-}E,fa}$ $\Delta pH$ = −1.19 vs. median $OMM^{12}$ $\Delta pH$ = −0.97 of the full community in AF medium, Fig. 2a). In the polysaccharide medium (APF), all communities exhibit more acidic spent culture supernatant pH (median $\Delta pH < -1.2$) compared to communities in the glucose medium (AF) with the exception of the *B. caecimuris* dropout consortium. Here, a less drastic change in pH was observed (median $OMM^{11\text{-}B,ca}$ $\Delta pH$ = −0.48), indicating that the strong acidification of the polysaccharide (APF) spent culture supernatant is predominantly due to the presence of *B. caecimuris*.

To test how the individual species changed their metabolic environment, we analyzed spent culture supernatants of all communities by untargeted metabolomics. All samples from communities lacking *E. faecalis* in the glucose condition (AF medium) and both, *B. caecimuris*, and *B. coccoides* in the polysaccharide condition (APF medium) showed distinct metabolomic profiles compared to the other communities (Fig. 2b, Fig. S3). Specifically, amino acid (AA) production and depletion profiles revealed that only communities including *E. faecalis* depleted serine and arginine in both culture conditions (Fig. 2c, d). On the other hand, communities lacking *B. caecimuris* in the polysaccharide condition (APF medium) stood out with lower levels of alanine compared to fresh medium, indicating the importance of this strain for alanine release. Targeted measurements of short-chain fatty acids (SCFA) levels in the spent culture media further revealed significantly decreased concentrations of acetic acid, propionic acid, succinic acid, isovaleric acid, isobutyric acid, and methylbutyric acid in *B. caecimuris* dropout communities cultured in the polysaccharide condition (APF medium, Fig. S4). In contrast, communities lacking *E. faecalis* cultured in the glucose condition (AF medium) had significantly increased levels of propionic acid, methylbutyric acid, valeric acid and isovaleric acid and significantly decreased lactic acid concentrations. Of note, butyric acid levels were strongly reduced in communities lacking *F. plautii*, a species previously shown to produce butyrate[35]. Consistently, butyric acid levels were also significantly decreased in the dropout community lacking *B. coccoides*, in which *F. plautii* was as well not detectable (Fig. S2).

**Species relationships are not transferrable across multiple different nutritional environments**

As community ecology and keystone species could not be transferred from one carbohydrate condition to the other, we aimed to elucidate strain relationships for the three identified environment-dependent keystone species in the other culture media as well (Table 2, Table S1). Therefore, the three dropout consortia lacking *E. faecalis*, *B. caecimuris* and *B. coccoides* were cultured in mGAM, TYG and YCFA media (Fig. S5).

To obtain a quantitative measure for strain relationships, we defined $r_{abs}$ as the ratio of the absolute abundance of a given strain y in a dropout community lacking strain x vs. the absolute abundance of strain y in the full consortium: $r_{abs} = \frac{abs_y^{OMM11-x}}{abs_y^{OMM12}}$. A strain relationship was defined as negative, if the abundance of a strain y was increased in a given dropout community, compared to the corresponding strain abundance in the full community ($r_{abs} > 1$). A strain relationship was defined as positive, if the abundance of a strain was decreased in a given dropout community, compared to the corresponding strain abundance in the full community ($r_{abs} < 1$). If a strain y was not detectable in a specific dropout community, but in the full community, the strain relationship was defined as a positive dependency. If a strain y was only detected in a specific dropout community, but never in the presence of the corresponding strain, the interaction outcome was defined as exclusion. This analysis revealed a strong variation of strain relationships across different culture media (Fig. 2e) including several cases of contrasting outcomes, e.g. the relationship between *E. faecalis* and *A. muciniphila*. Only three strain relationships were found to prevail across nutritional environments: the negative influence of *E. faecalis* on the species *B. animalis* and *C. inoccuum* and the positive relationship between *B. coccoides* and *M. intestinale*. The differential impact of *E. faecalis*, *B. caecimuris* and *B. coccoides* on strain interactions in different nutritional environments was once more reflected in their ability to alter the metabolic environment, as differences in pH modification were observed across all culture media (Fig. S6).

**E. faecalis interactions are multifaceted in the community context**

Next, we aimed to resolve mechanisms behind the observed strain relationships to gain insight into the context dependency of keystone interactions.

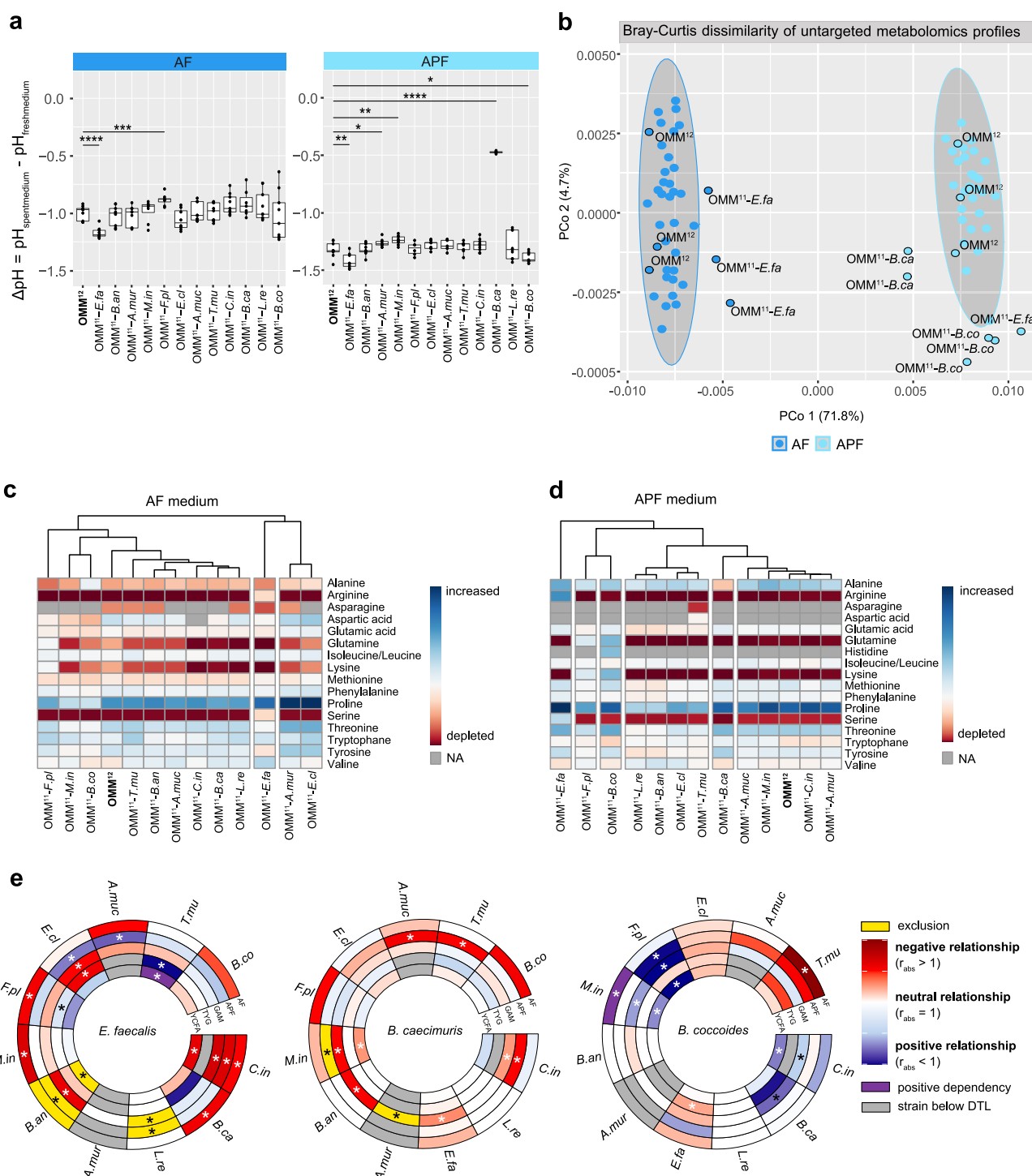

*E. faecalis* was previously shown to dominate strain relationships by metabolic interactions in a glucose-rich environment, but also to directly inhibit several other community members in co-culture (*B. animalis*, *F. plautii*, *E. clostridioformis*, *C. innocuum* and *L. reuteri*)[35]. *E. faecalis* harbors at least three bacteriocin-encoding loci[35], including enterocin L50, an enterococcal leaderless bacteriocin with broad target range among Gram-positive bacteria[40]. The inhibitory effect of *E. faecalis* in the community context was most pronounced for *B. animalis* (Fig. S5). By comparing bactericidal activity of the three individual *E. faecalis* mutants against the other OMM[12] members, we identified enterocin L50A and B as the active toxin (Fig. S7). Next, we analyzed the enterocin mutant strain *E. faecalis* ΔL50

in the community context in AF, APF and mGAM, to disentangle toxin mediated interactions from other modes of interaction (Fig. S8). *B. animalis* was abundant in the presence of *E. faecalis* ΔL50 but reduced in the presence of the wildtype in the glucose (AF medium) and polysaccharide condition (APF medium, Fig. 3a). Moreover, no significant difference in *B. animalis* loads was observed when *E. faecalis* ΔL50 condition was compared to the *E. faecalis* dropout community. This indicated, that interaction between *E. faecalis* and *B. animalis* in AF and APF is dominantly mediated by enterocin L50 (Fig. 3b). In mGAM however, the enterocin did not seem to play a role in the interaction of the two species: *B. animalis* abundance was only significantly increased in the *E. faecalis* dropout community but not when

**Fig. 2 | Context-dependent keystone species affect community assembly and strain relationships by environmental modification. a** pH modification of community cultures in AF and APF medium. ΔpH is shown as median (black line) with the corresponding upper and lower percentile (box, whiskers indicate 1.5 times interquartile range, points outside this range are considered outliers). Significant differences between OMM[12] ΔpH and OMM[11-x] ΔpH are depicted by asterisks ($N = 9$ each, two-sided $t$ test, $p$ values denoted as *<0.05, **<0.01, ***<0.005, ****<0.001, ns not shown). **b** Principle coordinate analysis of metabolomic profiles of community culture supernatants. Bray–Curtis dissimilarity analysis was performed on untargeted metabolomic profiles of the individual communities comparing the OMM[12] and the twelve OMM[11-x] communities ($N = 3$ each) in the two different culture media. Culture media are depicted in different colors. Gray ovals cluster each culture medium with a 95% confidence interval, outliers from the corresponding clusters, as well as the OMM[12] community for reference, are highlighted with a black ring. **c**, **d** Amino acid profiles of community culture supernatants in AF and APF medium ($N = 3$ each). Amino acids were annotated by

comparing the exact mass and MS2 fragmentation patterns of the measured features to the records in HMBD and the in-house database. Consumption of a specific amino acid from fresh medium is indicated in red, while production is indicated in blue. **e** Circular heatmaps depicting strain relationships inferred from changes in absolute strain abundances in the three dropout communities lacking *E. faecalis*, *B. caecimuris* and *B. coccoides*, respectively. Rings depict different culture media, colors indicate the relationship type as determined by $r_{abs}$: negative strain relationship in red, positive strain relationship in blue, exclusion in yellow (strain y only detected in a specific dropout community but never in the presence of the corresponding strain), positive dependency in purple (strain y not detectable in a specific dropout community but in the full community), absolute strain abundance under the detection limit (DTL) in gray. Significantly changed absolute abundances in the corresponding dropout communities compared to the full consortium are depicted by asterisks ($N = 9$ each, two-sided Wilcoxon test, $p$ values denoted as *<0.05). The corresponding data set is shown in Fig. S5. Source data are provided with this paper.

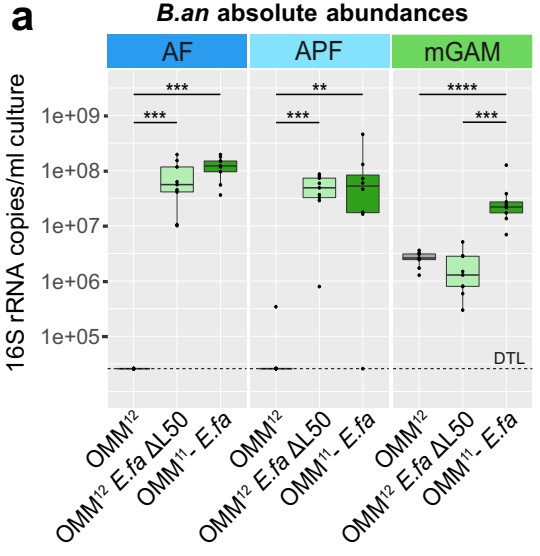

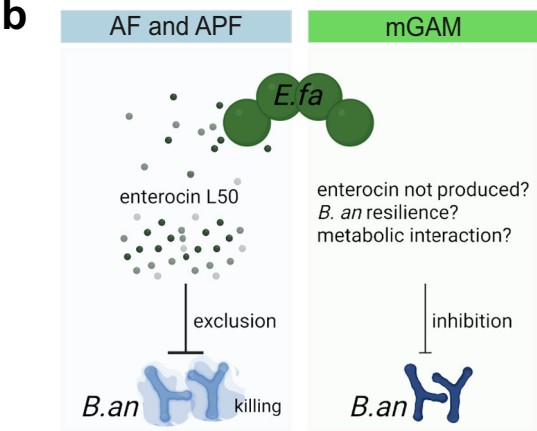

**Fig. 3 | Dissecting mechanisms underlying the context-dependent keystone function of *E. faecalis* in the community. a** Absolute abundances of *B. animalis* in the full consortium, the *E. faecalis* dropout community, and the OMM[12] community with a *E. faecalis* ΔL50 mutant strain grown in AF, APF, and mGAM media. Median (black line) absolute abundances are shown with the corresponding upper and lower percentile (box, whiskers indicate 1.5 times interquartile range, points outside this range are considered outliers). Significant differences between the different community cultures in the different culture media ($N = 9$ each) are depicted

by asterisks (two-sided Wilcoxon test, $p$ values denoted as *<0.05 **<0.01, ***<0.005, ****<0.0001, ns not shown). **b** Schematic overview of the interaction between *E. faecalis* and *B. animalis* in different cultivation media. While in AF and APF medium *B. animalis* is excluded from the community by *E. faecalis* enterocin L50 production, *B. animalis* is not affected by the enterocin in mGAM medium. The mechanism of interaction between *E. faecalis* and other community members is not solely explainable by interference or metabolic competition, but is likely multifaceted. Source data are provided with this paper.

the enterocin was deleted (*E. faecalis* ΔL50). This indicates that in mGAM, the enterocin is either not expressed, its effect is overpowered by a simultaneous compensatory interaction or *B. animalis* is insensitive to the toxin (Fig. 3a, b).

Interestingly, the interaction of *E. faecalis* with other community members previously affected in co-culture was similarly multifaceted. *C. innocuum* abundance was not significantly increased in the *E. faecalis* ΔL50 mutant community in all three tested medium conditions. This indicated that the observed negative effect of *E. faecalis* in the full consortium is not due enterocin production, but is mediated by other means such as substrate competition or end product inhibition (Fig. S8). The effect of the *E. faecalis* wildtype or the ΔL50 mutant strain on the abundance of *F. plautii* and *E. clostridioformis* was not consistent across the three different media, suggesting that the respective changes in absolute abundances of these species are not solely explainable by interference or metabolic competition. This highlights that exploitative and interference interactions can occur simultaneously and in a multifaceted fashion.

## The keystone function of *B. caecimuris* is dependent on the availability of inulin

The pronounced effect that *B. caecimuris* had on community assembly in the polysaccharide condition (APF medium, Fig. 1d) paired with the observation of strongly decreased spent culture medium pH (Fig. 2a) and altered metabolic profiles (Fig. 2b), suggested an interdependency between the availability of specific sugars or polysaccharides and the ability of *B. caecimuris* to alter this metabolic environment. This was substantiated by the presence of genes encoding the key enzymes of inulin and xylan degradation[41–43] in the *B. caecimuris* genome (Fig. 4a, Table S2). To analyze this further, we generated modified APF media with either none (APF[mod]) or only one of the two polysaccharides inulin and xylan (APF[inulin], APF[xylan]). Testing community assembly of the full consortium in the modified media revealed that the abundance of *B. caecimuris* was strongly increased by either inulin or xylan (Fig. S9).

In line with our hypothesis, in the absence of both polysaccharides none of the species initially affected by the absence of *B. caecimuris* (namely *M. intestinale*, *C. innocuum*, *T. muris* and *A. muciniphila*,

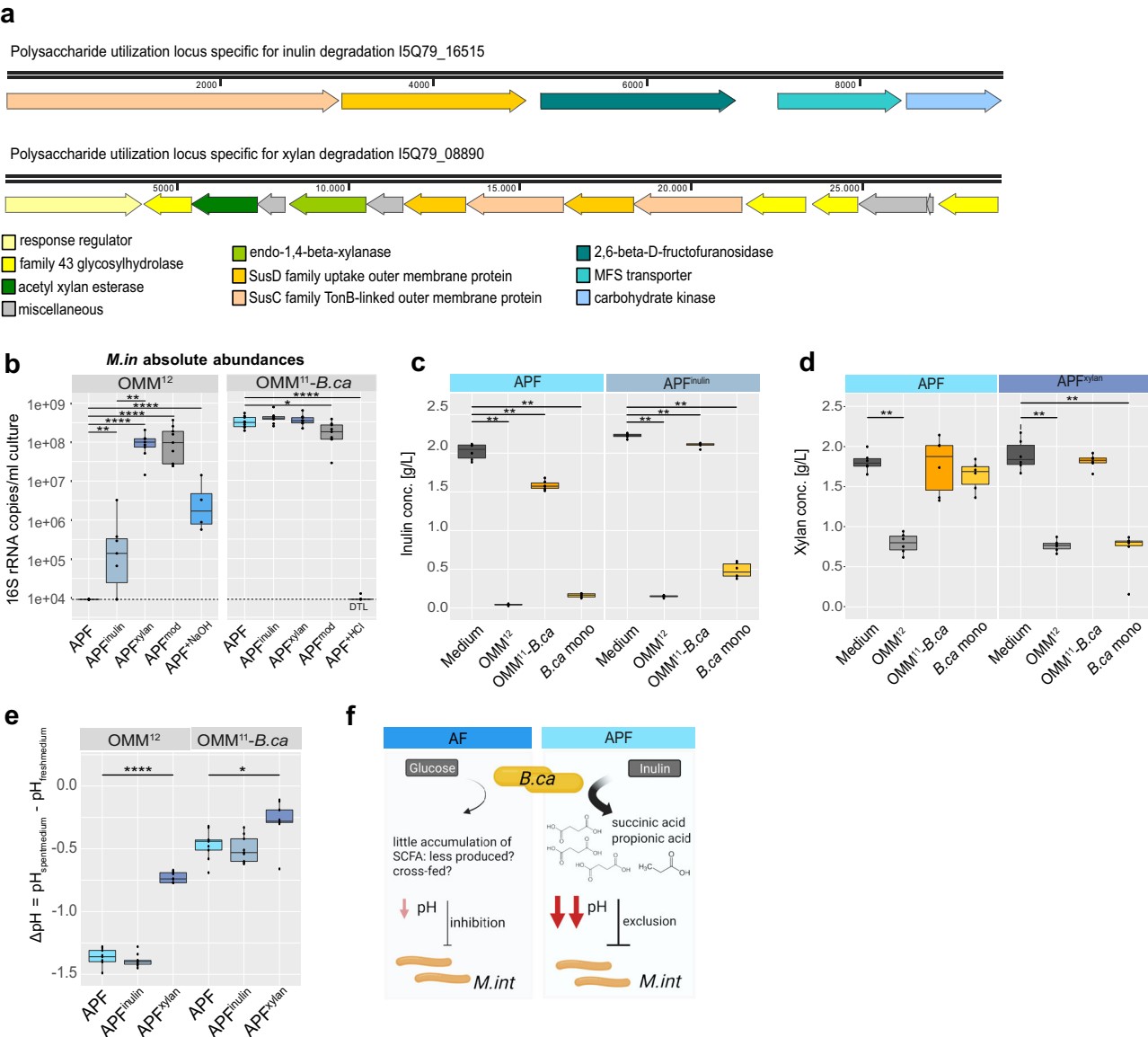

**Fig. 4 | Dissecting mechanisms underlying the context-dependent keystone function of *B. caecimuris* in the community. a** Polysaccharide utilization loci for xylan and inulin as identified in the genome of *B. caecimuris*. **b** Absolute abundances of *M. intestinale* in the full consortium and the *B. caecimuris* dropout community grown in modified APF media with both, no or single polysaccharides xylan (APF$^{xylan}$) and inulin (APF$^{inulin}$), as well as in APF media with adjusted pH (supplemented HCl or NaOH). Median absolute abundances (black line, $N = 9$ each) are shown with the corresponding upper and lower percentile (box, whiskers indicate 1.5 times interquartile range, points outside this range are considered outliers). Significant differences between the different culture media in the different community cultures are depicted by asterisks (two-sided Wilcoxon test, $p$ values denoted as *<0.05 **<0.01, ***<0.005, ****<0.0001, ns not shown). **c, d** Concentrations of inulin and xylan were measured using an enzymatic assay in fresh medium, and community spent culture supernatant of the full consortium, the *B. caecimuris* dropout community and *B. caecimuris* monocultures. Median concentrations (black line) are shown with the corresponding upper and lower

percentile (box, whiskers indicate 1.5 times interquartile range, points outside this range are considered outliers). Significant differences in different conditions are depicted by asterisks ($N = 3$ each, two-sided t-test, p values denoted as **<0.01, ns not shown). **e** pH modification of full community and *B. caecimuris* dropout community cultures grown in modified APF media. ΔpH is shown as median (black line) with the corresponding upper and lower percentile (box, whiskers indicate 1.5 times interquartile range, points outside this range are considered outliers). Significant differences between ΔpH of OMM$^{12}$ and the *B. caecimuris* dropout communities are depicted by asterisks ($N = 9$ each, two-sided *t* test, *p* values denoted as *<0.05, ****<0.001, ns not shown). **f** Schematic overview of the interaction between *B. caecimuris* and *M. intestinale* in different cultivation media. In the presence of inulin (APF medium), *B. caecimuris* is producing increased amounts of succinic acid and propionic acid. The resulting decrease in community culture pH leads to the exclusion of *M. intestinale*, a mechanism that is partially reversible by adjusting the culture pH. Source data are provided with this paper.

Fig. S5) showed significant changes in absolute abundances compared to the full consortium. This was especially clear for *M. intestinale* which was strongly reduced in the presence of *B. caecimuris* in the polysaccharide condition (APF medium) compared to APF lacking polysaccharides (APF$^{mod}$, Fig. 4b). This effect was more pronounced in the presence of only inulin compared to only xylan (log2 fold change of *M. intestinale* absolute abundance in the full vs. the dropout community

−9.44 vs. −1.85, respectively). This was reflected in the residual concentrations of inulin and xylan in the spent culture supernatants of *B. caecimuris* monocultures, *B. caecimuris* dropout communities and the full consortium. (Fig. 4c, d). While inulin was fully depleted (median conc$_{OMM12}$ = 0.04 g/L, median conc$_{B.ca}$ = 0.17 g/L in APF medium), xylan was only partially depleted by the community in the presence of *B. caecimuris* (conc$_{OMM12}$ = 0.79 g/L in APF medium) and in

*B. caecimuris* monocultures (median conc$_{B.ca}$ = 1.69 g/L in APF medium). Interestingly, xylan was only depleted in *B. caecimuris* monocultures when it was the only available polysaccharide. Correspondingly, changes in the spent culture supernatant pH were more pronounced in media containing inulin compared to media containing no polysaccharides or xylan only (Fig. 4e). This suggested that *B. caecimuris* produces certain fermentation products in the presence of inulin that strongly acidifies the culture pH and inhibit *M. intestinale* growth. This idea was supported by the observation that succinic and propionic acid were only significantly increased in the spent culture supernatant of communities including *B. caecimuris* in polysaccharide containing media (APF, Fig. S4) but not in media without polysaccharides (APF$^{mod}$ and AF medium, Fig. S10).

To test the effect of pH on community assembly independently of the metabolic end product, we chemically altered the pH of *B. caecimuris* dropout or full OMM[12] communities growing in APF medium (Fig. S11). By mimicking the previous observations, indeed *M. intestinale* could coexist with *B. caecimuris* in the alkalinized OMM[12] communities and was not detected in the acidified *B. caecimuris* dropout communities (Fig. 4b). Taken together, this suggests that capacity of *B. caecimuris* to exclude *M. intestinale* and consequently also its keystone role is mediated by culture acidification by the conversion of inulin to propionic and succinic acid (Fig. 4f).

### The role of *E. faecalis*, *B. caecimuris*, and *B. coccoides* in assembly and interactions in the murine intestinal tract

As *E. faecalis*, *B. caecimuris*, and *B. coccoides* were identified as important but context-dependent players in community assembly in in vitro conditions, we next set out to characterize their role in the gastrointestinal tract of stably colonized gnotobiotic mice. Germ-free mice (*N* = 8–10 mice per group) were colonized with the full OMM[12] consortium and dropout consortia lacking the three identified context-dependent keystone species *E. faecalis*, *B. caecimuris*, and *B. coccoides*, respectively. After 20 days of colonization, mice were sacrificed and different regions of the gastrointestinal tract (jejunum, ileum, cecum, colon, and feces) were sampled for qPCR analysis (Figs. S12, S13). Generally, lower bacterial loads (16S rRNA copies/g content) and higher variability across individual mice were found in the jejunum and ileum compared to cecum, colon and feces (Fig. 5a). Further, especially mice colonized with *B. caecimuris* and *B. coccoides* dropout communities showed reduced total bacterial loads in cecum, colon, and feces compared to mice colonized with the full consortium. Bray–Curtis dissimilarity analysis of community profiles, determined by the absolute abundances of the individual species across different sampling regions, revealed clear differences between community composition in the upper (jejunum and ileum) and in the lower gastrointestinal tract (cecum, colon and feces) (Fig. 5b). While none of the dropout communities stood out from the cluster of the upper gastrointestinal tract, the cecal community lacking *B. caecimuris* showed a distinct profile (Benjamini-Hochberg adjusted *p* value *p* = 0.0003, Table S3), indicating an influential role of this strain in the cecum.

Again, determining the measure r$_{abs}$ to quantify strain relationships from absolute strain abundances over the different gut environments revealed that *B. caecimuris* was mainly positively associated with most other species in the cecum (Fig. 5c, Fig. S13). This is in line with a particularly strong decrease in total bacterial abundance in the murine cecum in mice colonized with *B. caecimuris* dropout community (Fig. 5a). Comparing the relationship of *B. caecimuris* with the other OMM[12] species over the different gut regions revealed, again, contrasting relationships across the different sampling sites (Fig. 5c). Of note, quantifying bacterial strain relationships with *B. coccoides* revealed a strong positive dependency between *B. coccoides* and *F. plautii*, as the latter strain seems to be completely dependent on the presence of *B. coccoides* to establish itself in the murine gastrointestinal tract. A similar relationship, even though not as pronounced,

was observed for *C. innocuum*, which was as well positively associated with the presence of *B. coccoides* across all sampling sites. These two examples mark the only cases in which strain relationships were transferrable from the in vitro setup to the murine gut.

### Keystone function of *B. caecimuris* and *B. coccoides* is linked to altered intestinal physiology, inflammation, and metabolomics profiles in gnotobiotic mice

Mice colonized with communities lacking *B. caecimuris* and *B. coccoides* exhibited a significantly altered cecal-to-mouse body weight ratio (Fig. 6A). This ratio is known to be high in germ-free mice and strongly reduced in mice associated with a complex microbiota. For this reason, relative cecal weight is often used as an indicator of the impact of the microbiota on intestinal physiology. Corresponding to the strongly decreased total bacterial loads (Fig. 5a), the relative cecal weight was significantly increased in mice colonized with the *B. caecimuris* dropout community compared to mice colonized with the full consortium (Fig. 6a). In contrast, even though bacterial loads were decreased in mice colonized with the *B. coccoides* community, the relative cecal weight was strongly decreased in those mice compared to the full consortium control group. Of note, this effect was found to be caused by inflammation, characterized by crypt loss, neutrophil infiltration, and mucosal edema, as determined by histological assessment of the cecum (Fig. 6b, Fig. S14), as well as increased lipocalin-2 (LCN2) levels in the cecum of mice associated with the *B. coccoides* dropout community (Fig. 6c). The reason for this unexpected event of inflammation remains currently unclear. However, we observed this phenotype in several independent experiments.

Next, we analyzed intestinal samples of mice colonized with full and dropout communities using targeted and untargeted metabolomics. In line with the observation that the absence of species *B. caecimuris* and *B. coccoides* had a pronounced influence on overall bacterial loads, bacterial community assembly, and the host, strongly altered metabolomics profiles were observed for cecal content of mice colonized with communities lacking these species compared to the full consortium control (Fig. 6d). Targeted analysis of SCFAs (Fig. S15) and bile acids (Fig. S16) revealed pronounced differences across mice colonized with the different consortia. Specifically, a significant decrease in propionic acid, butyric acid, valeric acid, isovaleric acid, isobutyric acid, and 2-methylbutyric acid was found in the cecal content of mice colonized with the *B. caecimuris* dropout community compared to mice colonized with the full consortium. Contrasting, cecal content of mice colonized with communities lacking *B. coccoides* showed significantly increased levels of isovaleric acid and lactic acid, and significantly decreased levels of valeric acid and isobutyric acid compared to the full consortium control.

Concluding, we found that *B. caecimuris* and *B. coccoides* have a particularly strong influence on community assembly, total bacterial loads, and the metabolomic profile in the cecal content. Most strikingly, the important role of *B. coccoides* was not directly inferable from analyzing community composition only (Fig. 5b) but became apparent by analyzing inflammatory changes in the host and metabolomics profiles. Taken together, the effects of strain-dropouts include gut region-specific effects on microbial community assembly, changed total bacterial loads and metabolic profiles, and pronounced changes in host physiology.

### Discussion

Omics-based approaches have limited traceability of individual community members and experimental means to mechanistically interrogate interaction networks. The complementary use of synthetic communities provides a reductionist tool to mechanistically resolve interspecies interactions and identify keystone species[8,44]. Here, we explored the interaction network of the OMM[12] model community

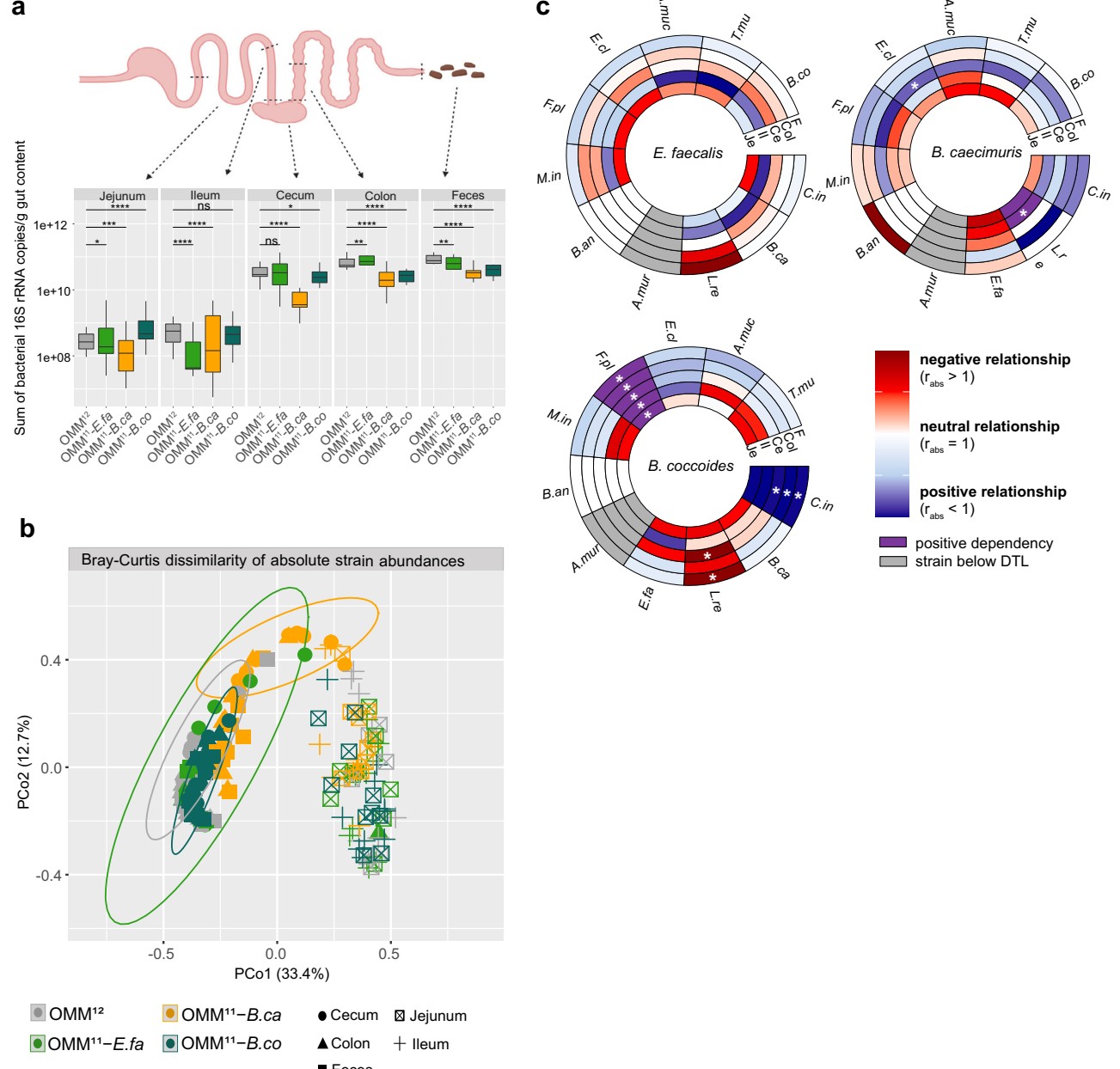

**Fig. 5 | Community assembly and interactions across different regions of the murine gastrointestinal tract. a** Germ-free C57Bl/6 J mice were inoculated with the full OMM[12] consortium and the three dropout communities OMM[11]-*E. faecalis*, OMM[11]-*B. caecimuris* and OMM[11]-*B. coccoides*, respectively (*N* = 8–10 mice per group). The median sum of absolute abundances (black line) are shown with the corresponding upper and lower percentile (box, whiskers indicate 1.5 times inter-quartile range) for all individual strains. Significantly changed sum absolute abundances in mice colonized with the corresponding dropout communities compared to mice colonized with the full consortium are depicted by asterisks (two-sided Wilcoxon test, *p* values denoted as ns = not significant, *<0.05 **<0.01, ***<0.005, ****<0.0001). **b** Principle coordinate analysis of community structure in different regions on the murine intestine. Bray–Curtis dissimilarity analysis was performed on absolute abundances of individual mice colonized with the OMM[12] or the three dropout communities lacking species *E. faecalis*, *B. caecimuris* and *B. coccoides* in the different regions of the mouse gut (*N* ≥ 8 mice per group mice per

group). Gut regions are shown in shapes, communities in different colors. Ovals cluster only samples from the murine cecum for each colonization (colors) with a 95% confidence interval. The corresponding statistical analysis is shown in Table S3. **c** Circular heatmaps depicting strain relationships inferred from changes in absolute strain abundances in the three dropout communities lacking *E. faecalis*, *B. caecimuris* and *B. coccoides*, respectively. Rings depict different sampling regions of the murine intestine, colors indicate the relationship type as determined by *r*abs: negative strain relationship in red, positive strain relationship in blue, positive dependency in purple (strain y not detectable in a specific dropout community but in the full community), absolute strain abundance under the detection limit (DTL) in gray. Significantly changed absolute abundances in the corresponding dropout communities compared to the full consortium are depicted by asterisks (*N* ≥ 8 mice per group, two-sided Wilcoxon test, *p* values denoted as *<0.05). The corresponding data set is shown in Fig. S13. Source data are provided with this paper.

---

using single-strain dropout communities across various culture media and in gnotobiotic mice. While we set out to determine the universal keystone species of this model gut bacterial community, we found that keystone roles are context-dependent[11,45,46]. Overall, we conclude that

true "keystone-ness" of a bacterial species that applies across different in vitro and in vivo conditions is rarely observed. Our data provides an experimental proof that keystone functions of focal microbes can differ in different nutritional and host-associated environments.

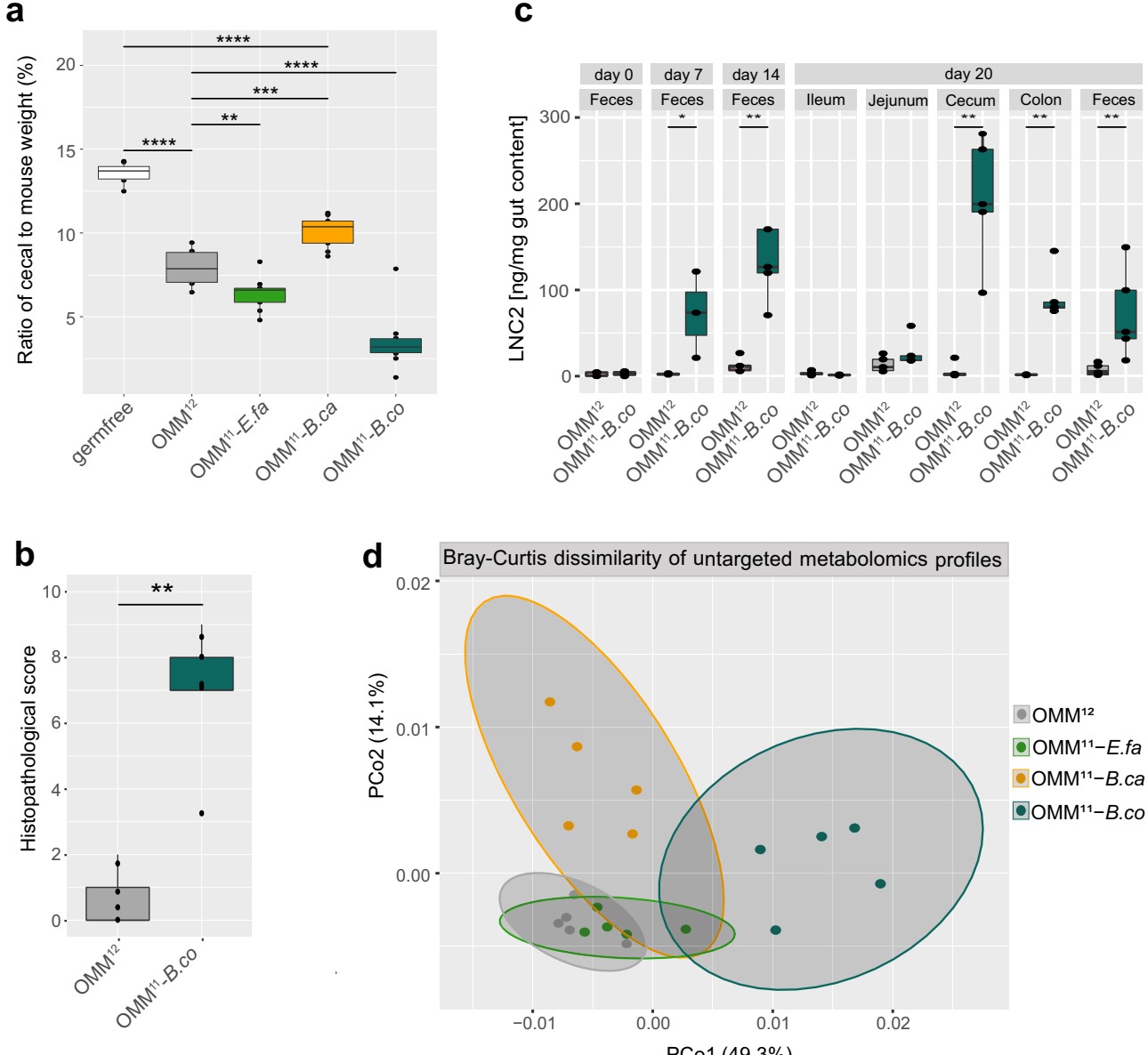

**Fig. 6 | Mice colonized with a *B. coccoides* dropout community exhibit gut inflammation and altered metabolomic profiles. a** Ratio of cecal to mouse body weight of mice colonized with the full consortium or the three dropout communities lacking *E. faecalis*, *B. caecimuris* and *B. coccoides*, respectively. Median cecal to mouse body weight ratios (black line) are shown with the corresponding upper and lower percentile (box, whiskers indicate 1.5 times interquartile range, points outside this range are considered outliers). Significantly changed ratios in mice colonized with the corresponding dropout communities compared to mice colonized with the full consortium are depicted by asterisks ($N \geq 8$ mice per group, two-sided t-test, p values denoted as *<0.05 **<0.01, ***<0.005, ****<0.0001). **b** Histopathological score of cecal tissue of mice colonized with the full consortium or the *B. coccoides* dropout consortium ($N = 5$ mice per group) is shown as median (black line) with the corresponding upper and lower percentile (box, whiskers indicate 1.5 times interquartile range, points outside this range are considered outliers). Significantly changed scores in mice colonized with the *B. coccoides* dropout community compared to mice colonized with the full consortium are

depicted by asterisks (t test, p values denoted as **<0.01). **c** Lipocalin LCN2 levels in the gut content of mice colonized with the full consortium and the *B. coccoides* dropout consortium over time ($N = 5$ mice per group) shown as median (black line) with the corresponding upper and lower percentile (box, whiskers indicate 1.5 times interquartile range, points outside this range are considered outliers). Significantly changed concentrations in mice colonized with the *B. coccoides* dropout community compared to mice colonized with the full consortium are depicted by asterisks (two-sided t test, p values denoted as *<0.05, **<0.01, ns not shown). **d** Principle coordinate analysis of metabolomic profiles cecal contents. Bray–Curtis dissimilarity analysis was performed on untargeted metabolomic profiles of cecal samples of mice colonized with the full consortium OMM[12] or the three dropout communities lacking *E. faecalis*, *B. caecimuris* and *B. coccoides* ($N = 5$ mice per group). Communities are depicted in different colors. Gray ovals cluster the corresponding colonization type with a 95% confidence interval. Source data are provided with this paper.

Specifically, we identified three environment-dependent keystone species, *E. faecalis*, *B. caecimuris,* and *B. coccoides*. Importantly, the kind and extent of how the three species affected community composition differed across culturing conditions and between sampling sites in the murine gut. While *E. faecalis* strongly influenced in vitro

community assembly by substrate competition and enterocin production, only minor changes in community structure, metabolomics profiles, and host physiology were observed in intestinal regions of gnotobiotic mice stably colonized with a *E. faecalis* dropout community. This difference could be due to the overall low relative absolute

abundance of *E. faecalis* in the gut of OMM[12] mice[35]. Increased abundance of *E. faecalis* is found in neonates[47], in antibiotic-treated individuals, graft-versus-host disease, or inflammatory bowel disease mouse models. It remains to be tested if *E. faecalis* takes on a keystone role under these conditions[48].

On the other hand, communities lacking *B. caecimuris* showed distinctly altered metabolomics profiles and differences in community assembly both in vitro and in the murine cecum. Notably, the presence of *B. caecimuris* was positively associated with most other species in vivo, whereas in vitro, more negative strain relationships were observed. *B. caecimuris'* keystone role in vitro was linked to the availability of inulin. *Bacteroides* spp. can break down polysaccharides by using surface-associated glycoside hydrolases[49,50], which can facilitate cross-feeding interactions with other community members[7]. Moreover, supplementing inulin as nutrient source boosts abundance of *Bacteroides* spp. and SCFA profiles in mice[42]. The metabolic exchange of polysaccharide fermentation products is thought to lead to a protective effect on intestinal cells[51]. These results suggest that exclusive nutrients enable specific species to take on a keystone role, a concept that might be generalized, but requires further experimental validation.

Most interestingly, the influential role of *B. coccoides* in vivo became especially apparent in hindsight of the strongly reduced cecal size and altered cecal metabolomics profiles. From analyzing community assembly in the murine cecum, *B. coccoides* would not have been inferred as species strongly affecting community structure. We hypothesize that the keystone role of *B. coccoides* is linked to higher-order interactions among other community members, as well as its important function as a hydrogen consumer in the OMM[12] community[35,52]. We reason that increased levels of hydrogen in the absence of this strain could alter the energy balance of hydrogen-producing fermentation reactions. In line with this, particularly butyrate producing species *C. innocuum* and *F. plautii* are strongly reduced in the absence of *B. coccoides*. Of note, mice associated with *B. coccoides* dropout community developed gut inflammation. Inflammation was detectable by increased fecal LCN2 already seven days after inoculation and symptoms aggravated until day 20 p.i. The reason for inflammation triggered by the community in the absence of *B. coccoides* is currently unknown. We speculate that other species may actively promote inflammation in the absence of *B. coccoides* (and *F. plautii*). The mechanisms underlying the role of *B. coccoides* in microbiota-host cross-talk will be the subject of future work, highlighting the challenges of transferring findings from an in vitro to an in vivo setting and vice versa.

Concluding, we and others discovered distinct differences in directionality and mechanisms in the underlying strain relationships across differing biotic and abiotic environmental conditions[53–55]. Reasons for the observed differences could be manifold, but most likely include the absence of host-derived factors, such as antimicrobial peptides[56], oxygen concentration gradients[57], and dynamic pH regulation[58] in the batch culture setup, as well as the lack of structural and spatial heterogeneity, that is present within the lumen of the gastrointestinal tract[59,60]. This indicates, that while similarly composed communities could assemble similarly across different environments, the underlying bacterial interaction networks and therefore resulting overall community functions might differ distinctly.

In summary, the presented insights into the conditionality and context dependency of bacterial interactions on the corresponding biotic and abiotic environment challenge the universal applicability and generality of the keystone species concept in the context of the gastrointestinal microbiome. We conclude that alterations in a community's interaction network may be overlooked by studying community composition and community-derived correlation only. Hence, systematic use of controllable community models, traceable

nutritional environments including chemically-defined cultivation media, and a combination of metagenomics and metabolomics approaches is needed to pave the way to elucidate the role of individual species in community functions and delineate general principles of how bacterial interactions shape microbiome function.

## Methods

### Generation of bacterial communities
Bacterial monocultures and subcultures were grown for 24 h each in 10 ml AF medium[35]. Dropout community inocula were prepared by first diluting the monocultures in fresh AF to OD600nm 0.1 (BioTek, Epoch2 Microplatereader) under anaerobic conditions. Species that had an OD600nm < 0.1 were used undiluted. For the generation of each dropout community inoculum, 500 µl of the monoculture dilution or the undiluted monoculture was mixed in a glass culture bottle. Accordingly, eleven different monocultures were used for each dropout community inoculum. The culture bottles with the dropout community inocula were hermetically sealed, discharged from the tent, filled into vials sealed with butyl rubber stoppers (10% glycerol), and frozen at −80 °C. Three biological replicates were generated from independently prepared monocultures in separately prepared batches of medium.

### Culture conditions for community experiments
Bacterial communities (three biological replicates with three technical replicates corresponding to in total nine wells) were cultivated in 24 well plates (TPP) under anaerobic conditions, thereby diluting the thawed inoculum 1:10 in 1 ml fresh media. Inocula were spiked with 100 µl fresh *M. intestinale* YL27 monoculture, to increase reliability of growth of this strain in the community. Bacterial communities were grown in five different media (Table S2): AF[35], APF (18.5 g l$^{-1}$ brain heart infusion glucose-free, 15 g l$^{-1}$ trypticase soy broth glucose-free, 5 g l$^{-1}$ yeast extract, 2.5 g l$^{-1}$ K$_2$HPO$_4$, 1 mg l$^{-1}$ haemin, 2.5 g l$^{-1}$ sugar mix (1:1 arabinose, fucose, lyxose, rhamnose, xylose), 2 g l$^{-1}$ inulin, 2 g l$^{-1}$ xylan, 0.025%.l$^{-1}$ mucin, 1 l dH$_2$O, 0.5 mg l$^{-1}$ menadione, 3% heat-inactivated fetal calf serum, 0.5 g l$^{-1}$ cysteine-HCl·H$_2$O, 0.4 g Na$_2$CO$_3$), modified GAM (Himedia Labs), TYG (10 g l$^{-1}$ tryptone peptone from casein, 5 g l$^{-1}$ yeast extract, 2 g l$^{-1}$ glucose, 0.5 g l$^{-1}$ L-cysteine-HCl, 100 ml l$^{-1}$ 1M K$_2$PO$_4$ pH 7.2, 40 ml l$^{-1}$ TYG salt solution (0.05 g MgSO$_4$−7H$_2$O, 1 g NaHCO$_3$, 0.2 g NaCl in 100 ml H$_2$O), 1 ml l$^{-1}$ CaCl$_2$ (0.8%), 1 ml l$^{-1}$ FeSO$_4$ (0.4 mg/ml), 1 ml hematine-histidine (pH 8, 0.2 M), 1 ml vitamin K3 (1 mg/ml$^{-1}$), 858 ml H$_2$O), YCFA (10 g l$^{-1}$ casitone, 2.5 g l$^{-1}$ yeast extract, 2 g l$^{-1}$ glucose, 2 g l$^{-1}$ starch, 2 g l$^{-1}$ cellobiose, 4 g l$^{-1}$ NaHCO$_3$, 1 g l$^{-1}$ L-cysteine, 0.45 g l$^{-1}$ K$_2$HPO$_4$, 0.45 g l$^{-1}$ KH$_2$PO$_4$, 0.9 g l$^{-1}$ NaCl, 0.09 g l$^{-1}$ MgSO$_4$·7H$_2$O, 0.09 g l$^{-1}$ CaCl$_2$, 10 mg l$^{-1}$ hemin, 1 l$^{-1}$ mg resazurin, 100 µl biotin (1 mg/10 ml in H$_2$O), 100 µl cobalamin (1 mg/10 ml in EtOH), 100 µl p-aminobenzoic acid (3 mg/10 ml in H$_2$O), 100 µl folic acid (5 mg/10 ml in DMSO), 100 µl pyridoxamin (15 mg/10 ml in H$_2$O), 1 l dH$_2$O, 100 µl Thiamine (5 mg/10 ml Stock in H$_2$O), 100 µl Riboflavin (5 mg/10 ml Stock in H$_2$O)). For the polysaccharide deficient variant APF$^{mod}$ of APF inulin, xylan and mucin were left out or either inulin (APF$^{inulin}$) or xylan (APF$^{xylan}$) were added in the initial concentration of 2 g/L. Over the total cultivation time of 96 h, 10 µl of the culture were transferred from one well into a new well with 1 ml of fresh medium every 24 h (1:100 dilution). For experiments testing the influence of pH on *M. intestinale* growth in the community, 10 µl NaOH (1 M) were added to the full consortium and 4 µl HCl (5 M) were added to the *B. caecimuris* dropout community each, 8 h after dilution every day to adjust the pH value of growing communities.

For sampling the 24-well plates were discharged from the tent and samples were rapidly processed. For each community culture, the full volume (1 ml) was centrifuged at 14,000 × *g* for 1 min, the cell pellet was frozen at −20 °C and the supernatant was kept for pH measurement and metabolomics analyses.

## pH measurements

pH measurements of bacterial supernatants were performed using a refillable, glass double junction electrode (Orion™ PerpHecT™ ROSS™, Thermo Scientific).

## Metabolomic profiling of bacterial supernatants and cecal content

For metabolomics, 500 µl supernatant or cecal content was transferred to a Spin-X centrifugation tube and centrifuged at $14{,}000 \times g$ for 2 min. The membrane was discarded and the flow-through snap-frozen in liquid nitrogen.

The untargeted analysis was performed using a Nexera UHPLC system (Shimadzu) coupled to a Q-TOF mass spectrometer (TripleTOF 6600, AB Sciex). Separation of the spent media was performed using a UPLC BEH Amide $2.1 \times 100$, 1.7 µm analytic column (Waters Corp.) with 400 µL/min flow rate. The mobile phase was 5 mM ammonium acetate in water (eluent A) and 5 mM ammonium acetate in acetonitrile/water (95/5, v/v) (eluent B). The gradient profile was 100% B from 0 to 1.5 min, 60% B at 8 min and 20% B at 10 min to 11.5 min and 100% B at 12 to 15 min. A volume of 5 µL per sample was injected. The autosampler was cooled to 10 °C and the column oven heated to 40 °C. Every tenth run a quality control (QC) sample which was pooled from all samples was injected. The spent media samples were measured in a randomized order. The samples have been measured in Information Dependent Acquisition (IDA) mode. MS settings in the positive mode were as follows: Gas 1 55, Gas 2 65, Curtain gas 35, Temperature 500 °C, Ion Spray Voltage 5500, declustering potential 80. The mass range of the TOF MS and MS/MS scans were 50–2000 m/z and the collision energy was ramped from 15–55 V. MS settings in the negative mode were as follows: Gas 1 55, Gas 2 65, Cur 35, Temperature 500 °C, Ion Spray Voltage −4500, declustering potential −80. The mass range of the TOF MS and MS/MS scans was 50–2000 m/z and the collision energy was ramped from −15 to −55 V.

The "msconvert" from ProteoWizard was used to convert raw files to mzXML (de-noised by centroid peaks). The bioconductor/R package xcms was used for data processing and feature identification. More specifically, the matched filter algorithm was used to identify peaks (full width at half maximum set to 7.5 s). Then the peaks were grouped into features using the "peak density" method[61]. The area under the peaks was integrated to represent the abundance of features. The retention time was adjusted based on the peak groups presented in most of the samples. To annotate possible metabolites to identified features, the exact mass and MS2 fragmentation pattern of the measured features were compared to the records in HMBD[62] and the public MS/MS database in MSDIAL[63], referred to as MS1 and MS2 annotation, respectively. The QC samples were used to control and remove the potential batch effect, $t$ test was used to compare the features' intensity from spent media with fresh media. For Bray–Curtis dissimilarity analysis of metabolomics profiles features with >80% NA values across the analyzed samples were removed. The associated untargeted metabolomics data are available on the MassIVE repository[64] with ID MSV000090704.

## Targeted short-chain fatty acid (SCFA) measurement

The 3-NPH method was used for the quantitation of SCFAs[65]. Briefly, 40 µL of the SM or cecal content and 15 µL of isotopically labeled standards (ca 50 µM) were mixed with 20 µL 120 mM EDC HCl-6% pyridine-solution and 20 µL of 200 mM 3-NPH HCL solution. After 30 min at 40 °C and shaking at 1000 rpm using an Eppendorf Thermomix (Eppendorf, Hamburg, Germany), 900 µL acetonitrile/water (50/50, v/v) was added. After centrifugation at 13,000 U/min for 2 min the clear supernatant was used for analysis. A Qtrap 5500 Qtrap mass spectrometer coupled to an Exion-LC (both Sciex) was used for the targeted analysis. The electrospray voltage was set to −4500 V, curtain gas to 35 psi, ion source gas 1 to 55, ion source gas 2 to 65, and the temperature to 500 °C. The MRM parameters were optimized using commercially available standards for the SCFAs. The chromatographic separation was performed on a $100 \times 2.1$ mm, 100 Å, 1.7 µm, Kinetex C18 column (Phenomenex, Aschaffenburg, Germany) column with 0.1% formic acid (eluent A) and 0.1% formic acid in acetonitrile (eluent B) as elution solvents. An injection volume of 1 µL and a flow rate of 0.4 mL/min was used. The gradient elution started at 23% B which was held for 3 min, afterward the concentration was increased to 30% B at 4 min, with another increase to 40% B at 6.5 min, at 7 min 100% B was used which was held for 1 min, at 8.5 min the column was equilibrated at starting conditions. The column oven was set to 40 °C and the autosampler to 15 °C. Data acquisition and instrumental control were performed with Analyst 1.7 software (Sciex, Darmstadt, Germany).

## Enzymatic assay to determine polysaccharide concentrations

Inulin was measured enzymatically using the Fructan HK Assay Kit (Megazyme Bray, Ireland) according to instructions. Samples were used directly, starting with protocol point E. Assay volume was reduced to half to allow measurements in 24-well plate format.

Xylan was measured enzymatically as xylose after acid hydrolysis using the D-Xylose Assay Kit (Megazyme Bray, Ireland) according to instructions. Xylan was hydrolyzed as described in Sample Preparation Example c) in the kit protocol and further measured in 24-well plates using the 3.75-fold amount of the given microplate assay procedure.

## Construction of exchange vectors for the engineering of *E. faecalis*

Vector pLT06 was used for the deletion of enterocins L50A-L50B, Ent96, and O16 in *E. faecalis* strain KB1[66]. A list of all the primers used is provided in Table S4. DNA fragments of 500–1000 bp upstream and downstream of the gene targeted for deletion (homologous arms 1 and 2) were amplified by PCR, using primers to insert restriction sites. *Bam*HI and *Sal*I restriction sites were added to arms 1, and *Sal*I and *Pst*I sites were added to arms 2. The PCR products of the arms were digested using the appropriate restriction enzymes and ligated with pLT06. The ligated products were transformed into NEB 10-beta Competent *E. coli* for propagation and grown on LB plates containing Chloramphenicol (Cm) at 30 °C. Colonies were screened for the presence of the inserts using primers pLT06_FW_b and pLT06_RV_b. Positive clones were grown overnight in liquid LB medium containing Cm at 30 °C. The plasmids were purified using the PureYield™ Plasmid Miniprep System (Promega) according to the manufacturer's protocol. The inserts from each construct were sequenced to ensure that no mutations arose during cloning. The resulting exchange vectors were used to generate the respective deletion mutants.

## Construction of *E. faecalis* enterocin mutant species

Deletion mutant species of *E. faecalis* KB1 were engineered by homologous recombination through double cross-over method, as described previously[66]. In brief, deletion exchanged vectors were transformed by electroporation into *E. faecalis* KB1. Transformed bacteria were grown on Brain Heart Infusion (BHI) agar plates containing Cm (20 µg/ml) and X-Gal (40 µg/ml) at 30 °C. Blue colonies were inoculated into 5.0 ml of Tryptic Soy Broth (TSB) containing Cm (20 µg/ml) and grown overnight at 30 °C. On the next day, the cultures were serially diluted and plated on BHI containing Cm and X-Gal and incubated overnight at 44 °C, to force single-site integration by homologous recombination. Light blue colonies were passaged by duplicating them on a BHI plate containing Cm and X-Gal, and then they were screened for the targeted integration using PCR with primers flanking the site of integration. Positive integration clones were grown overnight in TSB with no selection at 30 °C. On the next day, the cultures were serially diluted and plated on BHI containing X-Gal only and incubated overnight at 44 °C, to force the second site recombination event. The resulting white colonies were passaged by duplicating them on a BHI plate with no selection, and screened for the deletion of the target genes and loss of the plasmid by PCR. Genomic

DNA from colonies containing the deletions was amplified and sequenced to confirm gene deletions.

## DNA extraction and purification of in vitro and in vivo community samples

gDNA extraction using a phenol-chloroform-based protocol was performed as described previously[67]. First, three small spatula spoons of 0.1 mm zirconia/silica beads (Roth), 500 µl extraction buffer (200 mM Tris-HCl, 200 mM NaCl, 20 mM EDTA in ddH2O, pH 8, autoclaved), 210 µl 20% SDS and 500 µl phenol:chloroform:isoamylalcohol (lower phase) were added to the sample. Bacterial cells were lysed with the TissueLyser LT (Qiagen) set on max. speed (50 s-1) for 4 min. Following, samples were centrifuged at 5 min at full speed ($14,000 \times g$). The supernatant was transferred to new 1.5 ml tubes with 500 µl phenol:chloroform:isoamylalcohol. After mixing by inversion the samples were centrifuged again. The supernatant was transferred to new 2 ml tubes containing 1 ml 96% ethanol (p.a.) and 50 µl sodium acetate 3 M and mixed by inversion. The samples were centrifuged for min. 30 min at max. speed ($20,000 \times g$) at 4 °C. Subsequently, the supernatant was discarded. The pellet was resuspended in 500 µl ice-cold 70% ethanol, mixed by inversion and centrifuged 15 min at max. speed at 4 °C. The supernatant was discarded and the pellet was air dried for 5 min. For dissolving, the pellet was resuspended in 150 µl TE buffer (pH 8.0) and stored at 4 °C overnight. For gDNA purification, the NucleoSpin® gDNA Clean-up kit from Macherey-Nagel was used.

## Quantitative PCR of bacterial 16 S rRNA genes

Quantitative PCR was performed as described previously[36]. 5 ng gDNA was used as a template for qPCR. Strain-specific 16 S rRNA primers and hydrolysis probes were used for amplification. Standard curves were determined using linearized plasmids containing the 16 S rRNA gene sequence of the individual species. The standard specific efficiency was then used for absolute quantification of 16 S rRNA gene copies per 5 ng gDNA. Strain-specific detection limits (DTL) were used to exclude values below the DTL. Readouts were normalized for the strain-specific 16 S rRNA copy numbers, gDNA concentration of the corresponding sample, elution factor of the extracted gNDA, and sample volume.

## Genome screening for polysaccharide utilization loci in *B. caecimuris*

The genome of *B. caecimuris* I48 was screened for polysaccharide utilization loci (PUL) specific for inulin and xylan degradation found in literature[41,42] and a PUL database [http://www.cazy.org/PULDB/index.php?sp_name=Bacteroides+caecimuris+I48&sp_ncbi=]. Sequences of key enzymes for inulin and xylan degradation were blasted against the *B. caecimuris* I48 genome [https://www.ncbi.nlm.nih.gov/nuccore/CP065319] and names of key enzymes were checked in genome annotations of *B. caecimuris* I48 via word search ("1,4-beta-xylanase", "beta-xylosidase", "inulinase" and similar versions). Genome annotations around hits for key enzymes were screened for typical PUL structures (Table S1).

## Spot assays

Bacterial cultures and subcultures were grown for 24 h each in 10 ml AF medium at 37 °C under anaerobic conditions without shaking. Monocultures were diluted to $OD_{600nm}$ 0.1 in fresh AF medium. To generate a dense bacterial lawn, monoculture inocula were diluted in 3 ml LB soft agar to $OD_{600nm}$ 0.01 and poured on an AF medium agar plate. After drying all respective other bacteria were spotted onto the bacterial lawn in duplicates in a volume of 5 µl with $OD_{600nm}$ 0.1. Plates were incubated at 37 °C for 24 h under anaerobic conditions.

## Mice

All animal experiments were approved by the local authorities (Regierung von Oberbayern and Lower Saxony). Mice were housed under germ-

free conditions in flexible film isolators (North Kent Plastic Cages) or in Han-gnotocages (ZOONLAB) at a 12-hour light–dark cycle at $22 \pm 1.5$ °C and $50 \pm 5$ % humidity. The mice were supplied with autoclaved ddH2O and Mouse-Breeding complete feed for mice (Ssniff) ad libitum. For all experiments, female and male mice between 6–20 weeks were used and animals were randomly assigned to experimental groups. Mice were not single-housed and kept in groups of 2–6 mice/cage during the experiment. All animals were scored twice daily for their health status.

## Mouse experiments

Germ-free C57Bl/6 J mice were colonized with defined bacterial consortia (OMM[12], OMM[11]-*E. faecalis* KB1, OMM[11]-*B. caecimuris* I48, OMM[11]-*B. coccoides* YL58). Mice were inoculated as described previously[37]. Mice were inoculated two times (72 h apart) with the bacterial mixtures (frozen glycerol stocks) by gavage (50 µl orally, 100 µl rectally). All mice were sacrificed by cervical dislocation at 20 days after initial colonization. Intestinal content from ileum, cecum, colon, and feces were harvested, weighed, and frozen at −20 °C before DNA extraction. Cecal content was sampled in a 2 ml bead beater tube (FastPrep- Tubes Matrix D, MP Biomedical), weighed, snap-frozen in liquid nitrogen, and stored at −80 °C previous to metabolomics analyses.

## Hematoxylin and eosin staining

Cecal tissue, directly frozen in O.C.T was cut in 5 µm sections by using a cryotome (Leica) and mounted onto Superfrost Plus glass slides (Hartenstein). Sections were dried o. n. and fixed for 30 s in Wollman solution (95 % ethanol, 5 % acetic acid), washed in flowing tap water (1 min), and rinsed in dH2O. Afterwards, slides were incubated for 20 min in Vectors's Hämalaun (Roth) washed in flowing tap water (5 min), dipped one time in de-staining solution (70 % ethanol with 1 % HCl), washed again in flowing tap water (5 min) and rinsed in dH2O with subsequent rinses in 70 % and 90 % ethanol. Slides were then dipped for 15 s in alcoholic eosin (90 % ethanol) with Phloxin (Sigma-Aldrich), rinsed in dH2O followed by dehydration in 90 % ethanol, 100 % ethanol and xylene. Sections were directly mounted with Rotimount. Histopathological scoring of cecal tissue was performed as described previously[68]. Submucosal edema (0–3), infiltration of polymorphonuclear neutrophils (PMNs) (0–4), loss of goblet cells (0–3), and epithelial damage (0–3) were evaluated and all individual scores were summed up to give a final pathology score: 0–3 no inflammation; 4–8 mild inflammation; 9–13 profound inflammation.

## Lipocalin-2 quantification

Lipocalin-2 (LCN2) levels were determined by enzyme-linked immunosorbent assay (ELISA). The protocol was adapted from the Mouse Lipocalin-2/NGAL DuoSet ELISA kit from R&D Systems (Minneapolis, US). Briefly, ImmunoGrade™ 96-well plates (Brand) were coated with 50 µl lipocalin-2 capture antibody (1:200 in PBS) overnight at 4 °C in a humid chamber. Plates were washed three times in wash buffer (0.05 % Tween-20 in PBS) and blocked for one hour at RT with 100 µl blocking buffer (2 % BSA in PBS). Again, plates were washed three times and 50 µl of sample (undiluted (gut content in PBS) and 1:10 dilutions in PBS) were added in duplicates. For quantification, a standard with a starting concentration of 50 ng/ml lipocalin-2 was added and serially diluted 1:3 in blocking buffer. After 1 h of incubation at RT and six times washing, 50 µl of lipocalin-2 detection antibody (1:200 in blocking buffer) was added, subsequently incubated for 1 h at RT, and washed six times. With 100 µl of HRP-streptavidin (1:1,000 in PBS) the plate was incubated for 1 h at RT, six times washed, and developed 30 min at RT with 100 µl substrate (1 mg ABTS in 10 ml 0.1 M NaH2PO4 pH 4, 5 µl H2O2) in the dark. Absorbance was measured at $\lambda_{405}$.

## Statistical analysis

For comparison of absolute abundance levels of OMM[12] species between experiments, a two-sided Wilcoxon test was performed

using R Studio (version 1.2.5001). Obtained *p* values below $p = 0.05$ were considered statistically significant. The vegdist function of the R library *vegan* version 2.5–4 (https://github.com/vegandevs/vegan) was employed to obtain Bray–Curtis dissimilarities between the samples based on absolute abundances. Permutational multivariate analyses of variance (PERMANOVA) were performed in R using the function Adonis (method "bray" with 9999 permutations). Obtained p values were adjusted using the Benjamini–Hochberg method. In this study, no statistical method was used to predetermine sample size. No data were excluded from the analyses. The experiments were not randomized. Except for histopathological scoring, the investigators were not blinded to group allocation during experiments and outcome assessment.

## Data availability

The authors declare that all data supporting the findings of this study are available within the paper, its supplementary information, and its source data table files. Additionally, the untargeted metabolomics data generated in this study has been deposited in the MassIVE repository under accession ID MSV000090704 [https://doi.org/10.25345/C5222RB1N]. The *B. caecimuris* I48 genome is available under accession number CP065319. PUL were analyzed using the public PUL database cazy [http://www.cazy.org/PULDB/index.php?sp_name=Bacteroides+caecimuris+I48&sp_ncbi=]. The *E. faecalis* KB1 genome is available under accession number CP015410. Source data are provided with this paper.

## Code availability

All applicable software and codes used are stated and cited in the Methods along with information to operate the tools. The following software was used: Data were analyzed using R Studio (Version 1.2.5001). Heatmaps were generated using the R *pheatmap* package (https://github.com/raivokolde/pheatmap). Plots were generated using the R *ggplot2* package (https://ggplot2.tidyverse.org) and *ggpubR* package (https://github.com/kassambara/ggpubr). qPCR data were acquired using a Roche Diagnostics "LightCycler" with the corresponding software "LightCycler 96" (Version 1.1.0.1320). For targeted analysis of SCFA concentrations, data acquisition, and instrumental control were performed with Analyst 1.7 software (Sciex, Darmstadt, Germany). Untargeted metabolomics data was analyzed using the bioconductor/R package xcms was used for data processing and feature identification as described in detail in the methods section. Figures 1a, 3b, 4f and 5a were partly generated using BioRender (https://biorender.com). All Figures were arranged using Affinity Designer (Version 1.10.4.1198).

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

## Acknowledgements

The authors thank S. Hussain for excellent technical support and members of the Stecher laboratory for helpful feedback and discussions. We thank Giorgia Greter (ETH Zürich) and Carlos Geert Pieter Voogdt (EMBL, Heidelberg) for performing experiments that could not be included in the final version of the manuscript. This research received funding from the German Research Foundation (DFG, German Research

Foundation, Projektnummer 395357507—SFB 1371, Projektnummer 279971426 and 315980449, BS), the European Research Council (ERC) under the European Union's Horizon 2020 research and innovation program (Grant Agreement 865615, BS), the German Center for Infection Research (DZIF, BS) and the Center for Gastrointestinal Microbiome Research (CEGIMIR, BS).

## Author contributions

A.S.W. conceived and designed the experiments. A.S.W., L.S.N., A.v.S., A.G.B., and D.R. performed the experiments. A.S.W., L.S.N., C.M., and K.K. analyzed the data. C.M., K.K., C.L., and J.H. contributed materials, species, or analysis tools. A.S.W. and B.S. coordinated the project. A.S.W. and L.S.N. wrote the original draft and all authors reviewed and edited the draft manuscript.

## Funding

## Competing interests

The authors declare no competing interests.
