## [Peer Review File · Nature Communications]

REVIEWER COMMENTS

Reviewer #1 (Remarks to the Author):

Weiss et al constructed dropout communities and analyzed their structure and function in vitro and in the murine gastrointestinal tract. They found that certain species significantly affected community structure, but that the identity of these keystone species depended on growth media and location in the murine gut. This statement is important but it also seems obvious, e.g. species physiology and interspecies abundance correlations are known to vary with context. Importantly, what environmental differences affect interactions and how are not systematically addressed, or at least presented in a disorganized way which leaves the reader with a deluge of observations rather than an overall understanding. This issue is a major weakness.

That said, I think that the paper presents some interesting experimental results, including showing which species have a more significant impact on community composition in the OMM system, the dependency of the effect of *B. caecimuris* on the availability of polysaccharides, and the mouse results--which I think could be of interest to people working on the ecology of the gut microbiome. But the paper is quite dense, and I worry that it may not be readily accessible to people not already familiar with the OMM system.

Thus, my main suggestions are for the authors to a) tone down the message of biotic/abiotic context-dependency of community assembly; b) link their findings to other studies/model systems - which would also increase the generality of the findings and broaden the readership; and c) it would be helpful to add a few schematics with the environment- species metabolic interactions, at least for the 3 dropout communities studied in more detail.

Also, I find the heading "Dissecting strain-strain interactions in the community context using dropout consortia" (L151) a bit misleading. Here the authors are not really dissecting strain-strain interactions but rather showing species abundance correlations using single-species dropout communities. While correlations between 2 species may hint at interactions, it is not necessarily the case, and not demonstrated here. For instance, correlations between 2 species may be due to higher-order interactions. So I would rephrase the heading of the section.

Major comments:

- In general, the paper seems written for someone who is extremely familiar with this system. It is sometimes difficult to track the species by its current label, i.e., Fig 3A would be easier to digest if "148" was replaced with "B. caecimuris"
- To emphasize the overall message, it might be helpful to include a schematic map of the various environment-species interactions (or perhaps a network of environment-species-environment-species interactions). For example, in Fig. 4A, it might be useful to denote the environmental variables that shape the observed differences in interactions.
- Fig 1 C/D: It is not surprising that the two media produce two different spaces in the PC plot. I think the claim "that the two media produce different composition and metabolic profiles" is more of a supplemental figure/claim to me. The main figure should split the PC plot (re-do PC analysis for just-AF, then just APF, etc). For instance, OM11-KB1 look like outliers to the general AF bubble, but it could also be because the x-axis is so stretched to include the APF bubble. I would like to see if the outliers remain outliers in the new analysis, or if new outliers arise.
- Fig 2: It seems like some species are not present/stable in the full OM12 community (e.g., L. reuteri, A. muris, B. animalis, T. muris in AF). This makes it seem like in this set-up, not all 12 members are actually present, making it hard to actually compare if these are **true** drop-out communities." Perhaps it's how it's plotted, but the levels of those species ^ look the same as its abundance in the community it's dropped from. This is troubling because, if true, we are actually not comparing drop-out communities and rather random mixtures.
- Fig 4b: This is incomplete to me. It's problematic to identify these 3 species are "keystone species" in AF/APF media and then show that they change the pH in other media to show they act as "keystone" species too in these media. What the authors are not showing are the other pH measurements from all other OMM11 communities. I would not be surprised if other members are also changing the pH, and hence you could say they are just as important as these "keystone" species.

Minor comments:

- Line 104: it was unclear what AF and APF mean. There are a lot of acronyms to keep track in the paper, so maybe would be easier to just use names for the media that are more intuitive.
- Fig. 1B: The schematic makes it look like the culture/subculture/dropout consortia were cultured in tissue culture flasks?- Fig 1C-1D. I would highlight the OMM12 datapoint as a baseline reference.
- Fig. 2, plot for 148 in AF. On the x-axis, OMM11-KB1 should be highlighted in red as there is a significant change in strain abundance between the OMM11-KB1 and OMM12 communities. This is a dense figure, and a way to make it easier for the reader to hone in on the important points is to color-match the bar or add a shaded background to the red/blue columns.
- Figure 3A legend (L703). It reads "To quantify the environmental modification by community growth, the pH of the community spent medium (SM) was determined. ". pH is one dimension of

'environmental modification' - but there are many other dimensions. Also, while 2 spent media may be more dissimilar (e.g., in terms of metabolites overlap), their pH may be more similar compared to 2 spent media with greater metabolites overlap (as pH also depends on concentrations,..). Also, it is possible that the environment has been modified but the pH is buffered by the presence of other species (e.g. a species cross-feeding on the organic acids produced by another species). What the authors are measuring here is change in pH, so I would rephrase the legend's title to focus on that.

- Fig 3C: What does the dotted line represent and why is it changing in every case? Is the data shown for OMM12 and OMM11-KB1 the same as in Fig 2 or from a new set of experiments? If the former, what explains the discrepancy in abundances for *L. reuteri* I49 between Fig 2 and Fig 3? Clarify.

- Figure 4B legend (L734-737). Same comment as above.

Reviewer #2 (Remarks to the Author):

Dear editor,

I read with great interest the paper entitled "Nutritional and host environments determine community ecology and keystone species in a synthetic gut bacterial community". This study describes a bacterial model community called OMM consisting of twelve gut bacterial species, representing the major eubacterial phyla of murine gut microbiota. The study explores the interaction network of the OMM community by generating single-species dropout communities to identify keystone species driving community assembly. Authors found an environmental context-dependency and conditionality of the keystone species concept, highlighting the need for experimental validation of association networks in the relevant biotic and abiotic environment. This study is of interest because although microbial interaction and co-occurrence networks have been identified as powerful tools for understanding the underlying processes shaping the microbiome, there is still incomplete understanding of the mechanistic basis underlying these interactions. and synthetic model communities can be a powerful tool to enable systematic presence-absence studies and identification of keystone taxa that have a major influence on microbiome composition and function.

This study is well done, and interesting and will certainly contribute to advance the field of synthetic ecology.

Specific points

The paper is quite narrative. Adding number, statistics though the text would improve the understanding

OMM was designed to reflect functionality but ecological inference is limited to its representativity in the gut. Result might not reflect keystone species driving community assembly. The consortium does not seem to reflect typical murine gut microbiota. This may explain partially the different results in vitro and in vivo

Title: I would add mice in the title also not to make it too generic as the results may not reflect all ecosystems

Methods. What is the starting pH of the consortia. How do strains grow individually on the AF and AFP media?

Line 69: I would add previous papers using drop-out approaches

Line 71: Is reference 12 the good one for OMM?

Line 84: please add a bit more results of main messages of the previous in vitro study, especially for specific species

Line 101. It is mentioned that protocol differ from previous studies. Can you indicate in 1-2 lines why and the reason to transfer every 24h and how would the lack of transfer differentially impact the community? I find relevant to add figure S1B in main figure using quantitative barplot

How the experiments were repeated?

Line 104: Why AF and AFP were selected? If from previous papers, please mention it. Mention in rich (based on BHI). Add % of sugars in AFP

Line 108: mention which 4 strains are not detected. And why? This seems contradictory with the fact that this OMM can be stable in vitro.

What was the kinetic of OD, growth rate of the consortia?

Line 236: mention the media used and the general differences between them

Discussion

One limitation of rich media can be discussed and a next step involving chemically-defined media to precisely study the metabolites produced and cross-feeding

The study uses a small sample size of gnotobiotic mice, which may not be representative of the larger population and other mice strains.

PBP response to the reviewer's questions

General comments and explanation given for modifications in main and supplementary Figures

We would like to thank both reviewers for their critical but overall positive feedback and helpful comments and suggestions to revise the manuscript and improve its quality and impact. We substantially revised the manuscript accordingly in structure to improve its clarity and included a number of new experiments and datasets.

We have addressed all comments in our point-by-point response and revised the manuscript accordingly. Please find below a brief summary of the major changes, as well as two tables to document all modifications of Figures and Supplementary Figures.

To improve the manuscript's outline and make it more accessible also for readers not familiar with the synthetic model community (OMM¹²), we revised the overall structure by:

- clarifying the experimental approach, the description of the model community and our initial hypotheses
- including strain names rather than strain numbers by introducing a new letter-code to interpret the Figures more easily
- removing abbreviations
- moving OMM¹²- specific information not essential for the main argumentation of the manuscript to the supplemental information
- including new experiments to elucidate the mechanism of environment-dependent bacterial interactions
 - ➔ repetition of the batch culture experiment for the OMM¹² and *B. caecimuris* dropout community in APF medium (now the corresponding plots include 3 biological replicates with 3 technical replicates each instead of 2 biological replicates with 3 technical replicates in the previous version, increasing the strength of our statistical tests and confirming our initial findings)
 - ➔ testing OMM¹² and *B. caecimuris* dropout community growth in modified APF media with only inulin or xylan as supplemented polysaccharides, as well as in pH adjusted APF medium
 - ➔ Measurement of xylan and inulin concentrations in the spent culture medium of *B. caecimuris* monoculture, as well as OMM¹² and *B. caecimuris* dropout community cultures grown in APF, APF^{mod}, APF^{inulin}, APF^{xylan} medium
 - ➔ Measurement of succinic acid concentrations in the spent culture medium of all communities grown in AF, APF and APF^{mod} medium
 - ➔ Mouse experiment with mice colonized with the full or *B. caecimuris* dropout community to determine LCN2 levels in the feces/gut content over time, as well as histopathological assessment of the cecal tissue

Tab. 1: Modifications of the main Figures

New Figure Number		Previous Figure Number	Remark
Fig 1	A	adapted Fig. 1B	
	B	new	PCoA analysis of the community composition of the OMM ¹² community in five different cultivation media (based on absolute abundances)
	C	new	Absolute strain abundances of the of the OMM ¹² members in five different cultivation media
	D	Fig. 1C	
	E	Fig. S4B	
Fig 2	A	Fig. 1E	
	B	Fig. 1D	
	C	Fig. S3A	
	D	Fig. S3B	
	E	Fig. 4A	
Fig 3	A	Fig. 4C	
	B	new	Schematic overview of the proposed mechanism of interaction between E. faecalis and B. animalis in the community in different culture media
	C	Fig. S5B	
	D	new	Absolute abundance of M. intestinale in the community context in media containing single or no polysaccharides as well as in media with adjusted pH
	E	new, Fig. S5A adapted	Concentrations of inulin in the spent culture supernatant of different communities, as well as of B. caecimuris monoculture
	F	new, Fig. S5A adapted	Concentrations of xylan in the spent culture supernatant of different communities, as well as of B. caecimuris monoculture
	G	new, Fig. 3A adapted	pH differences of spent culture supernatants of different communities grown in media containing single or no polysaccharides
	H	new	Schematic overview of the proposed mechanism of interaction between B. caecimuris and M.intestinale in the community in different culture media
Fig. 4	A	Fig. 5A	
	B	Fig. 5B	
	C	Fig. 5C	
	D	Fig. 5D	
	E	new	Histopathological assessment of cecal tissue of mice colonized with the full OMM ¹² consortium or with communities lacking B. coccoides

	F	new	Lipocalin 2 levels in the gut content of mice colonized with the full OMM ¹² consortium or with communities lacking B. coccoides over time
	G	Fig. 5E	

Tab. 2: Modifications of the Supplementary Figures

New Figure Number	Previous Figure Number	Remark
Fig. S1 A Fig. S1 B	new	Relative strain abundances of the of the OMM ¹² members and spent culture supernatant pH differences of communities grown in five different cultivation media
Fig. S2	Fig. 2	
Fig. S3	Fig. S2	
Fig. S4	Adapted Fig. S3C	Addition of succinic acid measurements
Fig. S5	Fig. S9	
Fig. S6	Fig. 4B	
Fig. S7	Fig. S7	
Fig. S8	Fig. S8	
Fig. S9	new, adapted Fig. S5C	Absolute strain abundances of the of the OMM ¹² members in the full or B. caecimuris dropout community in media containing single or no polysaccharides
Fig. S10	Adapted Fig. S6/ Fig. 3B	Addition of succinic acid measurements
Fig. S11	new	Absolute strain abundances of the of the OMM ¹² members in the full or B. caecimuris dropout community in media with adjusted pH
Fig. S12	Fig. S10	
Fig. S13	Fig. S11	
Fig. S14	Fig. S12	
Fig. S15	Fig. S13	
Fig. S16	new	Bile acid concentrations in the cecal content of mice colonized with the full OMM ¹² community or with communities lacking E. faecalis KB1, B. caecimuris I48 or B. coccoides YL58

PBP to the reviewers comments

Reviewer #1 (Remarks to the Author):

Weiss et al constructed dropout communities and analyzed their structure and function in vitro and in the murine gastrointestinal tract. They found that certain species significantly affected community structure, but that the identity of these keystone species depended on growth media and location in the murine gut. This statement is important but it also seems obvious, e.g. species physiology and interspecies abundance correlations are known to vary with context. Importantly, what environmental differences affect interactions and how are not systematically addressed, or at least presented in a disorganized way which leaves the reader with a deluge of observations rather than an overall understanding. This issue is a major weakness.

We like to thank the reviewer for their constructive and helpful feedback on our work.

While we agree that the context-dependency and conditionality of bacterial interactions is initially intuitive, we would like to point out that the keystone species concept and terminology is still frequently used in gut microbiome literature without referring to its context-specificity. Several species have been termed keystones for the mammalian gastrointestinal system: species providing important metabolic functions as degrading polysaccharides (Ze et al., ISME J, 2012; Cartmell et al., Nat Microbiol, 2019; Centanni et al., Appl Environ Microbiol, 2018; Dapa et al., Cell Host Microbe, 2022) or producing short-chain fatty-acids (Tsukuda et al., ISME J, 2021), as well as species being associated with anti-inflammatory responses (Kropp et al., Sci Rep, 2021), restoring dysbiosis and promoting gut health (Wu et al., iMeta, 2022; Leylabadlo et al., Micro Pathog, 2020; Chng et al., Nat Eco Evo, 2020). In many cases, conditionality of these keystone functions is neglected, detailed description of the corresponding context is not provided or not achievable, and some studies even go as far as trying to identify disease or health associated keystone species that serve as *universal* biomarkers in medical assessments.

So far, systematic experimental testing of bacterial community ecology and the keystone species concept across different environmental contexts was rare. We think, that using a synthetic bacterial community model and a dropout community approach different common culture media in combination with gnotobiotic mouse models tackles an important knowledge gap in gut microbiome research. Our findings clearly highlight the pitfalls of assigning specific microorganisms fixed roles in the ecosystem and underline the difficulties in transferring observed bacterial interactions (pairwise or within a community) across different environmental contexts, including in vitro-in vitro, in vitro-in vivo and in vivo-in vivo comparisons. Hence, while intuitive, experimentally proving the context-dependency of bacterial interactions and the keystone species concept is - our opinion - novel, valuable and provides a potential perspective in future work on interpretation of microbiome signatures.

We agree with the reviewer that structure and clarity of the manuscript had to be improved. We revised the introduction to motivate and explain our hypotheses more clearly, removed data sets from the main text that were dispensable for the overall argumentation, organized our findings and included new experiments to systematically uncover which environmental differences affected the observed interactions in the community context. For example, we included several new experiments to systematically address the mechanism underlying *B. caecimuris* strong influence on community assembly in the presence of polysaccharides (new Fig. 3C-F). Thereby, we uncovered that by consuming inulin, *B. caecimuris* produced succinic acid and propionic acid in large quantities, which acidified the spent culture pH.

Testing the influence of pH-modification on community assembly (more acidic pH in the *B. caecimuris* dropout community, more alkaline pH in the full community compared to the corresponding controls) revealed that one of the strongest observed strain relationships (exclusion of *M. intestinales* by *B. caecimuris*) was mainly mediated by pH rather than metabolic exclusion or direct inhibition.

That said, I think that the paper presents some interesting experimental results, including showing which species have a more significant impact on community composition in the OMM system, the dependency of the effect of *B. caecimuris* on the availability of polysaccharides, and the mouse results--which I think could be of interest to people working on the ecology of the gut microbiome. But the paper is quite dense, and I worry that it may not be readily accessible to people not already familiar with the OMM system.

We agree that the initial version of the manuscript included a lot of information on the OMM system that was redundant to the main argumentation of the paper. We moved several large datasets (e.g. previous Fig. 2) from the main Figures to the supplementary information (see Overview Table 1 and 2 above). This way, the data is still available for the interested reader, while the argumentation in the main text is easier accessible and streamlined. Further, we introduced a letter-code to interpret Figures and keep track of the members of the community more easily, as suggested (new Tab. 1). We also removed abbreviations where possible to improve readability.

Thus, my main suggestions are for the authors to a) tone down the message of biotic/abiotic context-dependency of community assembly; b) link their findings to other studies/model systems - which would also increase the generality of the findings and broaden the readership; and c) it would be helpful to add a few schematics with the environment- species metabolic interactions, at least for the 3 dropout communities studied in more detail.

Summarizing the aforementioned points, in the revised version of the manuscript we streamlined the argumentation and improved the clarity of our initial hypotheses and used experimental approach, added links to related studies (a + b), as well as shortened and strengthened the identified environment-dependent bacterial interaction mechanisms and included schematic descriptions correspondingly (c). The schematic descriptions are included in the new Fig. 3B and H:

Also, I find the heading "Dissecting strain-strain interactions in the community context using dropout consortia" (L151) a bit misleading. Here the authors are not really dissecting strain-strain interactions but rather showing species abundance correlations using single-species dropout communities. While correlations between 2 species may hint at interactions, it is not necessarily the case, and not demonstrated here. For instance, correlations between 2 species may be due to higher-order interactions. So I would rephrase the heading of the section.

This paragraph (including the previous heading) was completely removed from the manuscript and the corresponding Figures was moved to the supplemental information. We agree that this formulation could have misled the reader as the used dropout community approach can only indicate interactions that arise within the community and not necessarily between two strains. We carefully formulated observed interaction patterns as "strain relationships" rather than strain interactions, to clearly indicate that linearity of interactions is not necessarily given (e.g. L172 and following in the revised version of the manuscript).

Major comments:

- In general, the paper seems written for someone who is extremely familiar with this system. It is sometimes difficult to track the species by its current label, i.e., Fig 3A would be easier to digest if "148" was replaced with "*B. caecimuris*"

We agree that the strain numbers are hard to follow for a reader not familiar with the OMM¹² community. Hence, we changed the Figure labels to a letter code (e.g. instead of KB1 for *E. faecalis* now *E.fa*) that is indicated in the first results section (new Tab. 1). Due to the density of some Figures we could not include the full strain names but think that the letter code is a compromise in keeping the Figures clearer.

- To emphasize the overall message, it might be helpful to include a schematic map of the various environment-species interactions (or perhaps a network of environment-species-environment-species interactions). For example, in Fig. 4A, it might be useful to denote the environmental variables that shape the observed differences in interactions.

We included two schematic depictions of the identified environment-dependent bacterial interactions of *E. faecalis* and *B. caecimuris* (see new Fig. 3B and H, as described above) and extensively revised the corresponding paragraphs. Further, we included several new experiments to strengthen the argumentation for the observed influence of *B. caecimuris* on community ecology in the presence of polysaccharides (as described above). The differences in environmental variables of the *in vitro* cultivation media are further now highlighted in the new Tab. 2 in the manuscript.

- Fig 1 C/D: It is not surprising that the two media produce two different spaces in the PC plot. I think the claim "that the two media produce different composition and metabolic profiles" is more of a supplemental figure/claim to me. The main figure should split the PC plot (re-do PC analysis for just-AF, then just APF, etc). For instance, OM11-KB1 look like outliers to the general AF bubble, but it could also be because the x-axis is so stretched to include the APF bubble. I would like to see if the outliers remain outliers in the new analysis, or if new outliers arise.

We initially included a PCo analysis as described (PCo analysis of community assembly of all communities in both media separately) but found that it led to the same result, as shown below. The same communities are found as outliers independent of the inclusion of just one or both media. To make the information readily accessible in one graph and also highlight that in most cases the strongest variability in community assembly does not come from the presence or absence of individual strains but from the culture media environment, we decided to include the PCo analysis as initially shown.

- Fig 2: It seems like some species are not present/stable in the full OMM12 community (e.g., *L. reuteri*, *A. muris*, *B. animalis*, *T. muris* in AF). This makes it seem like in this set-up, not all 12 members are actually present, making it hard to actually compare if these are *true* dropout communities." Perhaps it's how it's plotted, but the levels of those species ^ look the same as its abundance in the community it's dropped from. This is troubling because, if true, we are actually not comparing drop-out communities and rather random mixtures.

All community members are able to grow in at least one of the different media or one specific dropout community. Further, thorough checking of the community inocula and time-resolved community assembly in the early hours of the experiment (in contrast to community structure after four days of serial dilutions as shown in the manuscript) confirmed the reproducible presence of all twelve (or respectively eleven in the dropout communities) members. To understand the reasons why several species might not be detectable (hence, get excluded over time) in the full community context or the presence of specific strains in a given environment after community stabilization was a central focus of this study. Members of the community that are on many occasions not detectable after four days of serial dilutions are *L. reuteri* and *A. muris*. Both strains are detectable under specific conditions, e.g. *L. reuteri* in the absence of *E. faecalis* in APF medium and *A. muris* in the absence of *B. caecimuris* in mGAM (see Fig. S5 in the updated version of the manuscript). This highlights that in principle these strains are able to grow in a specific community context and environment.

As an example to show that dropout communities are indeed “real” dropout communities and differ from communities inoculated with all 12 members, one may appreciate e.g. the significant change in the culture supernatant pH of the *A. muris* dropout community in APF medium (see Fig. 2A in the updated version of the manuscript), or the significant increase in *E. faecalis* abundance in the *B. animalis* and *L. reuteri* dropout communities in AF and APF medium, respectively (see Fig. S2 in the updated version of the manuscript). This indicates that indeed the initial presence of these strains (even though being excluded in the full

consortium over the course of four days) can affect community dynamics. In summary, we are confident that the studied communities are indeed “true” dropout communities and carefully tested our experimental setup to provide controllability and reproducibility of our approach.

- Fig 4b: This is incomplete to me. It's problematic to identify these 3 species are "keystone species" in AF/APF media and then show that they change the pH in other media to show they act as "keystone" species too in these media. What the authors are not showing are the other pH measurements from all other OMM11 communities. I would not be surprised if other members are also changing the pH, and hence you could say they are just as important as these "keystone" species.

Unfortunately, while of great interest, it was not feasible to test community structure, pH, metabolic profiles etc. of all thirteen communities across five medium conditions. The described experimental approach is quite labor-intensive and providing the full data set for all medium conditions was beyond our capacities.

Still, we agree that the initial structure of the manuscript misleadingly implied that we assumed that only the species identified in AF and APF medium act as keystone species. We fully agree that it is very likely that we would have identified other members of the community as keystone species in the other media conditions. Hence, we carefully revised the structure of the manuscript highlighting that we were rather interested in how the bacterial strain relationships and community ecology changed in other media conditions with a special focus on the previously identified keystone members (see L174-178 in the revised version of the manuscript).

Minor comments:

- Line 104: it was unclear what AF and APF mean. There are a lot of acronyms to keep track in the paper, so maybe would be easier to just use names for the media that are more intuitive.

We reduced the number of acronyms and abbreviations in the main text and revised the main text thoroughly to explicitly describe the corresponding conditions, e.g. L132 in the revised version of the manuscript and several other instances. Further, as described above, we included a new Tab. 2 in the manuscript, listing and describing the composition of the media used in the study.

- Fig. 1B: The schematic makes it look like the culture/subculture/dropout consortia were cultured in tissue culture flasks?

Indeed, the pre-cultures that precede mixing of the community inocula were generated in tissue culture flasks. This allows culturing bacterial communities in the anaerobic tent in a higher surface/volume ratio to facilitate gas exchange. Community inocula were assembled in serum bottles and stored in cryo-vials. Community batch culture experiments were then performed in 24-well plates.

- Fig 1C-1D. I would highlight the OMM12 datapoint as a baseline reference.

We agree that this is a helpful addition to the plot. We highlighted the OMM¹² datapoints as a reference in Fig. 1D and 2B.

- Fig. 2, plot for I48 in AF. On the x-axis, OMM11-KB1 should be highlighted in red as there is a significant change in strain abundance between the OMM11-KB1 and OMM12 communities. This is a dense figure, and a way to make it easier for the reader to hone in on the important points is to color-match the bar or add a shaded background to the red/blue columns.

Yes, we corrected and revised the Figure for better understanding. We agree that this Figure is very dense and does not add to the main argumentation of the manuscript and hence moved it to the supplementary information (new Fig. S2).

- Figure 3A legend (L703). It reads "To quantify the environmental modification by community growth, the pH of the community spent medium (SM) was determined. ". pH is one dimension of 'environmental modification' - but there are many other dimensions. Also, while 2 spent media may be more dissimilar (e.g., in terms of metabolites overlap), their pH may be more similar compared to 2 spent media with greater metabolites overlap (as pH also depends on concentrations..). Also, it is possible that the environment has been modified but the pH is buffered by the presence of other species (e.g. a species cross-feeding on the organic acids produced by another species). What the authors are measuring here is change in pH, so I would rephrase the legend's title to focus on that.

We fully agree with the reviewer and recognize that our previous formulation did not satisfy the complexity of the possible underlying mechanisms. Hence, we changed the corresponding Figure legends (e.g. L718 and L777 in the revised manuscript).

- Fig 3C: What does the dotted line represent and why is it changing in every case? Is the data shown for OMM12 and OMM11-KB1 the same as in Fig 2 or from a new set of experiments? If the former, what explains the discrepancy in abundances for *L. reuteri* I49 between Fig 2 and Fig 3? Clarify.

The dotted line in the absolute abundance plots indicates the detection limit (DTL) of the used approach. Absolute strain abundances were determined by strain specific real-time PCR (qPCR). Therefore strain-specific 16S rRNA primers and hydrolysis probes were used for amplification and standard curves were determined using linearized plasmids containing the 16S rRNA gene sequence of the individual species. The standard specific efficiency was then used for absolute quantification of 16S rRNA gene copy numbers of individual species (as described in the method section L584 and following in the revised version of the manuscript). The readout of this method correspondingly gives the number of 16S rRNA gene copies in 5ng extracted gDNA (the used template).

Hence, this method has a fixed strain specific detection limit, the lowest number of 16S rRNA gene copies that can still be detected in a sample of 5 ng gDNA, which we use to exclude readouts (replace with NA) that are below the corresponding DTL in the first step of analysis. Therefore, in the downstream analysis (where the copy number is normalized to the gDNA concentration, elution volume of the extracted gDNA (50 µl) and sampled volume, e.g. 1 ml culture or 1 g intestinal content/feces) all data is above the strain specific DTL per definition or set to NA. But to be able to include samples that gave a readout below the strain specific

DTL in the plots, we needed to include a normalized strain specific DTL as well. This step proposes the difficulty of deciding how the fixed strain specific DTL (per 5 ng gDNA) should be normalized. We want to emphasize that this is why we apply the DTL in the first step of analysis before normalization of the readout to make sure only confident data are included in the analysis. Only in the last step of analysis, the values previously set to NA (that were below the strain specific DTL per 5 ng gDNA) were set to the normalized DTL to be included in the plot.

In the initial version of the manuscript, we normalized the fixed strain specific DTL for every individual plot, by calculating a “plot” specific DTL with the lowest gDNA concentration and the corresponding volume shown. This led to changing DTLs between plots and hence data appearing differently in the plots, while being the same (as e.g. in previous Fig. 2 and 3 for *L. reuteri* as outlined by the reviewer, as in Fig. 3 additional data of the *E. faecalis* enterocin mutant community was added and led to a higher plot specific DTL).

Therefore, in the revised version of the manuscript, we decided to calculate one strain-specific DTL for all in vitro experiments and all in vivo experiments, each with the overall lowest measured gDNA concentration and normalized to the elution factor and sample volume of 1 ml culture / 1 g intestinal content, and included this in all plots. We agree that the previously chosen approach was misleading and unclear. We further added a corresponding explanation to the Methods section (L584 and following in the revised version of the manuscript).

- Figure 4B legend (L734-737). Same comment as above.

Changed, as described above.

Reviewer #2 (Remarks to the Author):

Dear editor,

I read with great interest the paper entitled “Nutritional and host environments determine community ecology and keystone species in a synthetic gut bacterial community”. This study describes a bacterial model community called OMM consisting of twelve gut bacterial species, representing the major eubacterial phyla of murine gut microbiota. The study explores the interaction network of the OMM community by generating single-species dropout communities to identify keystone species driving community assembly. Authors found an environmental context-dependency and conditionality of the keystone species concept, highlighting the need for experimental validation of association networks in the relevant biotic and abiotic environment. This study is of interest because although microbial interaction and co-occurrence networks have been identified as powerful tools for understanding the underlying processes shaping the microbiome, there is still incomplete understanding of the mechanistic basis underlying these interactions. and synthetic model communities can be a powerful tool to enable systematic presence-absence studies and identification of keystone taxa that have a major influence on microbiome composition and function.

This study is well done, and interesting and will certainly contribute to advance the field of synthetic ecology.

We thank the reviewer for their encouraging comments on our work!

Specific points

The paper is quite narrative. Adding number, statistics though the text would improve the understanding

We generally streamlined and shortened the main text to put more strength to our argumentation and facilitate easier reading of the manuscript. While doing so, we also included absolute values or statistical analysis readouts where they were helpful to interpret the Figures (e.g. L146, L253 in the revised version of the manuscript and several other instances).

OMM was designed to reflect functionality but ecological inference is limited to its representativity in the gut. Result might not reflect keystone species driving community assembly. The consortium does not seem to reflect typical murine gut microbiota. This may explain partially the different results *in vitro* and *in vivo*

Indeed, the OMM¹² community was initially designed to mirror the phylogenetic and functional diversity of the murine gastrointestinal tract, while still providing a synthetic community of intermediate diversity. The twelve members of the community were isolated from the murine cecum and were selected to represent the five major phyla present in the murine gastrointestinal tract. Previous to this and as well as the preceding study (Weiss et al., 2022, ISME J), this model community has been primarily used in the mouse model and has proofed itself a valuable tool to study functions of the gut microbiome *in vivo* (as e.g. colonization resistance against invading enteropathogens). Now and in the previous study (Weiss et al., 2022, ISME J), we provided a comprehensive analysis of community ecology *in vitro* and designed several experimental approaches to study strain behavior, interactions, and community assembly *in vitro* to complement the existing rich set of *in vivo* experiments. This makes the OMM¹² model a rare example of a synthetic bacterial community that was extensively studied in different *in vivo* contexts, as well as *in vitro*.

While we agree, that a twelve-member community can impossibly reflect all facets of a natural murine gut microbiome, we think that the OMM community is a very well-suited model system to test context-dependency of bacterial ecology both *in vitro* and *in vivo*. The observed differences between *in vitro* and *in vivo* dynamics are likely due to the difficulties in designing an *in vitro* setup that adequately reflects the *in vivo* scenario, not necessarily due to the choice of the community model. We emphasized these points and elaborated the corresponding part in the discussion (L397 and following in the revised version of the manuscript) to highlight possible reasons for differences between the *in vitro* and *in vivo* settings and tried to outline the limitations of the used model.

Title: I would add mice in the title also not to make it too generic as the results may not reflect all ecosystems

We fully agree, that our specific findings will not be transferrable to all ecosystems – indeed a central point of the main argumentation of the manuscript. We therefore tried to reflect this message in the title: different environments (biotic and abiotic *in vitro* as well as *in vivo*) alter community dynamics and keystone functions. By specifically indicating in the title that a synthetic community model was used, we would argue that it becomes clear that this study does not refer to the human microbiota. Further, the use of a murine gut community model

and the specific experimental outline is indicated in the abstract (L27, L31 in the revised version of the manuscript).

Methods. What is the starting pH of the consortia. How do strains grow individually on the AF and AFP media?

The starting pH of all media is around pH 7 with slight deviations between batches and media of around pH 0.1-0.2. Absolute values are indicated in the provided source data table. To track the actual modification of pH rather than medium/batch specific deviations we analysed the Δ pH, the difference between spent culture medium pH and fresh medium pH, as described in L116-117 in the revised version of the manuscript. Growth behavior and pH modification in monoculture of the OMM¹² strains was previously described in detail (Weiss et al., ISME J, 2022).

Line 69: I would add previous papers using drop-out approaches

A (due to the limitation in possible citations) not complete list of papers using dropout community approaches is included in the revised version of the manuscript (see e.g. references 30 and 33).

Line 71: Is reference 12 the good one for OMM?

The indicated '12' in L71 of the previous version of the manuscript does not indicate a reference, but belongs to the synthetic community model's name: OMM¹²

Line 84: please add a bit more results of main messages of the previous in vitro study, especially for specific species

While we agree that it would be both useful and interesting to connect the findings of the presented study to our previous work on the OMM¹² model in more detail, we decided to rather focus on the general ecological question: how do different biotic and abiotic environments affect community ecology and keystone species/functions? Hence, in the main text of the manuscript the OMM¹² community is mainly described as the model facilitating the investigation of our hypotheses rather than aiming for a specific description of this model system. Responding to the comments of Reviewer 1, we aimed to make our findings accessible for a broader readership that is not necessarily familiar with the used model community. Still, we tried to include as much information as possible for the reader specifically interested in the OMM¹² model community and further added all full data sets to the supplementary information as well as provided all data in the raw data table.

Line 101. It is mentioned that protocol differ from previous studies. Can you indicate in 1-2 lines why and the reason to transfer every 24h and how would the lack of transfer differentially impact the community?

The community batch culture protocol mainly differs in two factors: previously we propagated the communities for full 10 days with serial dilutions every 24 h. In the presented study, we propagate the communities for 4 days with serial dilutions every 24 h. Secondly, in

the previous study, community inocula were mixed from freshly grown pre-cultures and community batch cultures were directly inoculated. Due to the increased workload in preparing the dropout communities, we decided to pre-mix the inocula from freshly grown pre-cultures and store the inocula in cryo-vials at -80°C before inoculating the community batch cultures. The differences in the protocol lead to different outcomes in the final community structure, but overall trends remain the same. Due to practicability and efficiency we decided to use the adapted protocols as described. The batch culture method is described in the manuscript (L100-103 in the revised version of the manuscript) and a short description of alterations to the previous protocol was included in the supplementary information (Fig. S1).

I find relevant to add figure S1B in main figure using quantitative barplot

We changed the overall structure of the first results section and now included absolute abundance data and a PCo analysis of community composition of the full community in different cultivation media to the new main Figure 1 (Fig. 1 B, C) to provide the reader with a better understanding of the OMM¹² reference data set to interpret the results on dropout communities. We decided to keep the relative abundance bar plots in the supplementary information, as absolute values paired with the community structure analysis shown in the PCo plot provide the needed information in a more quantitative way, whereas relative abundance barplots e.g. cannot account for changes in the overall bacterial abundances (e.g. TYG medium leads to generally lower bacterial loads).

How the experiments were repeated?

We assume the reviewer refers to the community batch culture experiments. Experiments were repeated as described in the methods section (e.g. Methods L426 and L430 and following in the revised version of the manuscript) and in the Figure legends. In short, we constructed three independent biological replicates of all community batch cultures (including independently constructed inocula and freshly cooked batches of culture media). From each biological replicate three technical replicates were generated in one experimental run, hence leading to a total of 9 replicates for each community batch culture. In some instances, individual replicates of dropout communities had to be excluded due to contaminations (with the initially excluded species) that occurred during the experiments. Hence, all experiments included at least 6, but mostly 9 replicates.

Line 104: Why AF and AFP were selected? If from previous papers, please mention it. Mention in rich (based on BHI). Add % of sugars in AFP

AF and APF medium were used in our preceding study on the ecology of the OMM community (Weiss et al., 2022, ISME J). The source and overall composition of the media used in the presented study is now indicated in Tab. 2 in the main text. Detailed recipes with exact carbohydrate concentrations of media are indicated in the methods sections (L434 and following in the revised version of the manuscript) and in Tab. S2 (same as previous Tab. S2).

Line 108: mention which 4 strains are not detected. And why? This seems contradictory with the fact that this OMM can be stable in vitro.

All community members are able to grow in at least one of the different media or one specific dropout community. To understand the reasons why several species might not be detectable (hence, get excluded over time) in the full community context or in the presence of specific strains in a given environment after community stabilization was a central focus of this study. Members of the community that are on many occasions not detectable after four days of serial dilutions are *L. reuteri* and *A. muris*. Both strains are detectable under specific conditions, e.g. *L. reuteri* in the absence of *E. faecalis* in APF medium and *A. muris* in the absence of *B. caecimuris* in mGAM. This highlights that in principle these strains are able to grow in a specific community context and environment.

Further, community stability does not necessarily imply that all members initially present in the inoculum will be present after stabilization, but rather refers to a stable community composition that does not change drastically anymore over time.

By restructuring the manuscript's outline we tried to clarify the characteristics of the reference data set of the OMM¹² community in all five media at the beginning of the manuscript (L93 and following, as well as L122 and following in the revised version of the manuscript). Thereby, we hope to provide the reader with the corresponding context for the following dropout community experiments. This includes that several species might be lacking from the full consortium in a given cultivation medium, motivating the study of dropout communities to potentially understand the reason of the exclusion of the corresponding strains.

What was the kinetic of OD, growth rate of the consortia?

Details on the growth kinetics of the OMM¹² members as well as the community batch cultures was previously described in Weiss et al. (2022, ISME J). Even though the batch culture protocol was slightly modified to the previous description (see above, indicated in the Figure legend of Fig. S1), overall growth rate and OD was comparable to previous measurements. Hence, measurements of OD and growth rate of communities were not included in the chosen approach.

Line 236: mention the media used and the general differences between them

We included a new Tab. 2 to the main text of the manuscript to outline the major differences between the media used in this study immediately in the beginning. Further, we the initial manuscript already included a detailed table of medium compositions in Tab. S1, as well as a description of medium composition in the methods section (L434 and following), to make medium composition transparent and accessible.

Discussion

One limitation of rich media can be discussed and a next step involving chemically-defined media to precisely study the metabolites produced and cross-feeding

We agree that this is an important point for the discussion of the presented study. We added a corresponding sentence to the discussion (L411 in the revised version of the manuscript).

The study uses a small sample size of gnotobiotic mice, which may not be representative of the larger population and other mice strains.

We assume that the reviewer refers to the use of mice colonized with the OMM12 or OMM dropout communities only, rather than testing other gnotobiotic mouse models (e.g. mice colonized with the ASF, SPF mice or transgenic mouse lines). The overall number of mice used per experiment was 8-10 individuals per group. The used number of mice refers to the number of mice that is stated in the ethical approval and was calculated to have sufficient power to detect biologically relevant changes.

While we agree that it would be interesting to test community assembly of other commonly used bacterial model communities or community assembly in different genetic mouse backgrounds the experiments are quite time and cost intensive and including different model organisms would have gone beyond the scope of the presented study. We would argue that the main points of the study (transferability of community ecology *in vitro* to *in vivo*; influence of selected members of community assembly *in vivo*) are sufficiently outlined by using the OMM¹² model in the context of gnotobiotic C57Bl/6J mice.

REVIEWERS' COMMENTS

Reviewer #1 (Remarks to the Author):

The revised manuscript is much improved. The writing is much clearer, and the main finding stands out: Species interactions are context-dependent, including interactions involving several "keystones". The authors also carried out new experiments to investigate the mechanisms behind how these interactions change in different environments, and the findings are summarized well by schematics. I have no other comments and recommend publication.

Reviewer #2 (Remarks to the Author):

Dear authors,

I have read the revised version and I have no further comments. Congratulations for the work.